# Child Health, Agriculture and Integrated Nutrition (CHAIN): protocol for a randomised controlled trial of improved infant and young child feeding in rural Zimbabwe

Laura E Smith,[1] Dexter. T Chagwena [iD],[2,3] Claire Bourke,[4] Ruairi Robertson [iD],[4] Shamiso Fernando,[2] Naume V Tavengwa,[2] Jill Cairns,[5] Thokozile Ndhlela,[5] Exhibit Matumbu,[2] Tim Brown,[6] Kavita Datta,[6] Batsirai Mutasa,[2] Alice Tengende,[2] Dzivaidzo Chidhanguro,[2] Lisa Langhaug,[2] Maggie Makanza,[2] Bernard Chasekwa,[2] Kuda Mutasa,[7] Jonathan Swann,[8] Paul Kelly [iD],[9] Robert Ntozini [iD],[10] Andrew Prendergast[6]

For numbered affiliations see end of article.

**Correspondence to**
Dexter. T Chagwena;
tungadex@gmail.com; d.
chagwena@zvitambo.com

## ABSTRACT

**Introduction** Over one-quarter of children in sub-Saharan Africa are stunted; however, commercial supplements only partially meet child nutrient requirements, cannot be sustainably produced, and do not resolve physiological barriers to adequate nutrition (eg, inflammation, microbiome dysbiosis and metabolic dysfunction). Redesigning current infant and young child feeding (IYCF) interventions using locally available foods to improve intake, uptake and utilisation of nutrients could ameliorate underlying pathogenic pathways and improve infant growth during the critical period of complementary feeding, to reduce the global burden of stunting.

**Methods and analysis** Child Health Agriculture Integrated Nutrition is an open-label, individual household randomised trial comparing the effects of IYCF versus 'IYCF-plus' on nutrient intake during infancy. The IYCF intervention comprises behaviour change modules to promote infant nutrition delivered by community health workers, plus small-quantity lipid-based nutrient supplements from 6 to 12 months of age which previously reduced stunting at 18 months of age by ~20% in rural Zimbabwe. The 'IYCF-plus' intervention provides these components plus powdered NUA-45 biofortified sugar beans, whole egg powder, moringa leaf powder and provitamin A maize. The trial will enrol 192 infants between 5 and 6 months of age in Shurugwi district, Zimbabwe. Research nurses will collect data plus blood, urine and stool samples at baseline (5–6 months of age) and endline (9–11 months of age). The primary outcome is energy intake, measured by multipass 24-hour dietary recall at 9–11 months of age. Secondary outcomes include nutrient intake, anthropometry and haemoglobin concentration. Nested laboratory substudies will evaluate the gut microbiome, environmental enteric dysfunction, metabolic phenotypes and innate immune function. Qualitative substudies will explore the acceptability and feasibility of the IYCF-plus intervention among participants and community stakeholders, and the effects of migration on food production and consumption.

## STRENGTHS AND LIMITATIONS OF THIS STUDY

⇒ Individually randomised trial to demonstrate impact of a new proposed infant and young child feeding (IYCF) and behaviour change intervention.
⇒ Community-based study using local nutrient-dense foods conducted in a rural Zimbabwean community.
⇒ Improved IYCF practices promoted through a behaviour change strategy at the onset of complementary feeding period between 5 and 6 months.
⇒ Measurement of a broad range of biomarkers including dietary intake, anthropometry and laboratory assays.
⇒ Limitation of a short follow-up period to measure outcomes.

**Ethics and dissemination** This trial is registered at ClinicalTrials.gov (NCT04874688) and was approved by the Medical Research Council of Zimbabwe (MRCZ/A/2679) with the final version 1.4 approved on 20 August 2021, following additional amendments. Dissemination of trial results will be conducted through the Community Engagement Advisory Board in the study district and through national-level platforms.

**Trial registration number** NCT04874688.

## INTRODUCTION

Undernutrition underlies 45% of child deaths among children <5 years[1]. Linear growth failure in childhood is the most prevalent form of undernutrition globally. An estimated 149 million children under 5 years of age are stunted, with a length-for-age Z-score (LAZ) more than 2 SD below the population median.[2] Stunting affects almost one-third of children in sub-Saharan Africa, leading to reduced human capacity and increased

long-term risk of chronic disease; it is therefore a surrogate marker of child health inequalities.[1]

The period from 6 to 24 months of age is one of the most critical phases of linear growth,[3] when stunting prevalence peaks due to high demand for nutrients coupled with limited quality and quantity of complementary foods.[2] Infant diets in rural sub-Saharan Africa often have low dietary diversity and a heavy reliance on white maize, which is high in starch and low in other nutrients. Interventions to improve infant and young child feeding (IYCF) typically include nutrition counselling to caregivers, plus a combination of commercial and locally available food products with or without micronutrients. However, a meta-analysis[4] of 42 studies showed only a modest impact of complementary feeding interventions on linear growth. Small-quantity lipid-based nutrient supplements (SQ-LNS), which are micronutrient-fortified ready-to-use products, show a small but measurable impact on LAZ.

We recently conducted the Sanitation Hygiene Infant Nutrition Efficacy (SHINE) trial,[5] a 2×2 factorial cluster-randomised trial of improved water, sanitation and hygiene and improved IYCF in rural Zimbabwe. A combination of IYCF messages and provision of SQ-LNS between 6 and 18 months of age improved LAZ of children at age 18 months by+ 0.16 (95% CI 0.08 to 0.23) and reduced stunting by 20%.[5] The intervention also increased haemoglobin by 2.0 g/L (95% CI 1.3 to 2.8) and reduced anaemia by almost 25%.[5] However, despite this intensive IYCF intervention, 32%, 73% and 23% of infants did not meet energy, folate and zinc/iron dietary intake requirements, respectively, as determined by 24-hour recall in a subgroup at 12 months and over one-quarter remained stunted. We also found evidence for barriers to infant nutrition, including caregiver capabilities,[6] household characteristics,[5] infant enteropathogen carriage,[7] and systemic and intestinal inflammation,[8 9] which were not resolved by the IYCF intervention.[10] Thus, we believe that persistent barriers to nutrient intake, uptake and utilisation limited the impact of the IYCF intervention.

Nutrient intake is influenced by food insecurity, household purchasing power and women's disempowerment to make decisions about land use, crop choice and distribution of food within the household; and inequitable gender beliefs.[5 11] Nutrient uptake and utilisation are influenced by intestinal pathologies which are highly prevalent among children in low-resource settings.[3] First, environmental enteric dysfunction (EED), a subclinical pathology of the small intestine characterised by intestinal inflammation and blunted villi, may impair efficient intestinal uptake of nutrients. Second, disturbance of the normal assembly of the gut microbiota may impair its roles in immune maturation, intestinal development and nutrient metabolism, thereby impairing growth.[12] Third, systemic inflammation arising from gut pathology increases energy requirements, reduces circulating micronutrients and inhibits the growth hormone axis.[13] Previously, barriers to intake, uptake and utilisation of nutrients have largely been addressed in isolation;

however, addressing these in parallel could ultimately improve growth and development in young children. Here, we present methodology for the Child Health Agriculture Integrated Nutrition (CHAIN) trial, which aims to address each of these barriers together through a randomised IYCF intervention.

## STUDY OVERVIEW

CHAIN is an open-label, individually randomised household trial comparing the effects of IYCF versus an enhanced IYCF intervention ('IYCF-plus') on energy and nutrient intake, growth and haemoglobin in infants at high risk of stunting. The trial was completed in February 2022 while the trial design paper was under review. The overarching goal of this trial is to fill key nutrient gaps among infants in rural sub-Saharan Africa through an improved IYCF intervention using locally available foods that could ultimately be sustainable through agriculture. A total of 192 rural Zimbabwean children were enrolled in the trial between 5 and 6 months of age; interventions were delivered from 6 to 12 months of age and the primary endline outcome of energy intake was assessed at 9 months of age (window 9–11 months). Interventions continued to be delivered until 12 months of age regardless of whether the endline visit was already completed, and all children were followed for endline visits to obtain the primary outcome. Our approach builds on the SHINE IYCF package, which reduced stunting but did not close all nutrient gaps.[5] We expect the CHAIN population will be similar to the SHINE population described above, as they are from the same rural community. CHAIN will test the impact of additional foods (powdered NUA-45 bio-fortified sugar beans, whole egg powder, moringa leaf powder and provitamin A (PVA) maize) that are nutrient-rich, culturally acceptable, locally sustainable and may have functional properties to ameliorate underlying pathogenic pathways, thereby tackling the identified barriers to nutrient intake, uptake and utilisation. For the duration of the trial, these foods were provided by community health workers (CHWs) as dried powders, which can be added to infant porridge as point-of-use fortificants. However, if shown to be efficacious, this trial would provide strong proof-of-principle that a comprehensive improvement to complementary feeding using locally available foods can substantially improve child nutrition. The chosen intervention foods have potential for local communities ultimately to become self-sufficient through modifications and adaptations to local agricultural systems that include local production and processing of these foods.

## STUDY OBJECTIVES
### Objective 1
Evaluate the effect of an enhanced infant feeding intervention ('IYCF-plus') on energy intake at 9 months of age (window 9–11 months) in a randomised, community-based trial in rural Zimbabwe. We hypothesise that

provision of powdered fortificants (PVA maize, NUA45 sugar beans, moringa and egg) for infants from 6 months of age will provide more energy at 9 months of age than the current standard-of-care IYCF intervention (trial primary outcome).

## Objective 2

Evaluate the impact of IYCF-plus on nutrient intake, growth and haemoglobin at 9 months of age (window 9–11 months) in a randomised, community-based trial in rural Zimbabwe. We hypothesise that IYCF-plus will improve the intake of key nutrients (protein, iron, zinc and folate) in 9-month-old infants compared with the standard-of-care IYCF intervention, and that IYCF-plus will increase length-for-age, weight-for-age, weight-for-length and haemoglobin more than IYCF (all secondary outcomes).

## Objective 3

Evaluate the impact of IYCF-plus on biological barriers to nutrient uptake and utilisation at 9 months of age (window 9–11 months) in a randomised, community-based trial in rural Zimbabwe. We hypothesise that the IYCF-plus intervention will increase microbiota maturity, ameliorate EED, reduce systemic inflammation and improve innate immune function in children aged 9 months, compared with the standard-of-care IYCF intervention.

## Objective 4

Identify metabolic signatures of the IYCF-plus intervention in young children at 9 months of age (window 9–11 months) in a randomised, community-based trial in rural Zimbabwe. We hypothesise that the IYCF-plus intervention will increase the concentrations of essential amino acids and choline at 9 months of age more than the standard-of-care IYCF intervention.

## Objective 5

Explore the acceptability and feasibility of the IYCF-plus intervention among participants and community stakeholders using qualitative methodology. Information from this assessment will be shared with policy makers to help design a larger roll-out of this intervention at district, provincial or national level.

## Objective 6

Explore the extent to which women's empowerment influences IYCF practices and nutrition outcomes in rural smallholder agricultural households.

We hypothesise that infants of women scoring in the highest tertile of the Women's Empowerment Agriculture Index (WEAI) will have improved macronutrient and micronutrient intake compared with infants of women in the lowest tertile of the WEAI index.

## Objective 7

Identify the extent of regional and international migration and movement (both rural–rural and rural–urban) at the household level, explore the type, frequency and impact of any associated remittance flows on food consumption and production, and consider the importance of migration to any changes in established food cultures. Information from this assessment will be shared with policy makers to help design a larger roll-out of this intervention at district, provincial or national level.

## RATIONALE FOR INTERVENTIONS

The CHAIN trial will compare IYCF as tested in the SHINE trial[5] vs an enhanced IYCF intervention ('IYCF-plus'). IYCF comprises a set of sequential behaviour change modules focusing on improved IYCF practices (eg, nutrient density, feeding during illness and dietary diversity), together with provision of daily SQ-LNS from 6 to 12 months of age, and powdered maize to make infant porridge. IYCF-plus comprises all the components of the IYCF intervention, plus four additional food supplements: PVA maize, NUA-45 sugar beans, moringa leaf powder and whole egg powder. We have chosen this combination of 'functional' food supplements to close the remaining nutrient gaps for young children identified during SHINE (table 1) and to ameliorate pathogenic pathways that impede uptake and utilisation of nutrients.

PVA Maize is a biofortified maize rich in beta-carotenes, which is grown in Zimbabwe. Studies in neighbouring countries have shown that daily intake of PVA maize can improve the vitamin A status of children.[14–18] PVA maize appears less prone to contamination with aflatoxin, which is a fungal toxin affecting agricultural crops during growth, storage and processing that may impair child growth.[19]

NUA45 sugar beans are a high-nutrient bean variety providing bio-fortified zinc and iron, a high-protein efficiency ratio, plus folate and resistant starch. Biofortified beans significantly increased haemoglobin, serum ferritin and body iron in African populations including Rwandan women.[20–23]

Moringa oleifera is a widespread crop in Zimbabwe. Dried moringa leaves can be ground into a powder providing a rich source of protein, fibre, mineral and micronutrients, including vitamin A, calcium, folate, vitamin C and vitamin E, and antioxidant polyphenols. Moringa leaf powder is available in shops in Zimbabwe as a food supplement. Pilot studies show that moringa leaf powder is safe and widely accepted as a dietary supplement by children and caregivers in sub-Saharan Africa,[24 25] but there have been no randomised trials.

Whole egg powder is commercially available, easily reconstituted and retains the nutrient content of whole eggs. Egg production is common in rural Zimbabwe. One egg per day for 6 months to children between 6–15 months of age increased LAZ by 0.63 in Ecuador.[26] This large effect is likely attributable to high-quality protein and choline, which are critical nutrients for linear growth.[27 28] Eggs contain all nine essential amino acids in proportions that closely match infant requirements for organ and muscle mass accretion. Choline is an essential

**Table 1** Nutrient provision in IYCF and IYCF-plus trial arms

Figure 1. Nutrient requirements from complementary food and provided by IYCF and IYCF-PLUS intervention diets with and without estimated mealie meal consumption for three age groups of children. Red (fails to meet requirement), orange (meets >88% requirement), green (exceeds requirement).*

| Nutrient | 6–8 months Required from complementary food | IYCF | IYCF-PLUS | 9–11 months Required from complementary food | IYCF | IYCF-PLUS | 12–24 months Required from complementary food | IYCF | IYCF-PLUS |
|---|---|---|---|---|---|---|---|---|---|
| Energy (kcal) from supplement alone | 270 | 118 | 222 | 451 | 118 | 247 | 746 | 118 | 272 |
| Energy (kcal) from supplement+estimated mealie meal intake | | 243 | 347 | | 368 | 497 | | 618 | 772 |
| Protein (g) | 7.7 | 2.6 | 10.8 | 8.1 | 2.6 | 12.6 | 8.1 | 2.6 | 14.4 |
| Fat (g) | 0.8 | 9.6 | 15.7 | 7.6 | 9.6 | 15.8 | 18.0 | 9.6 | 16.0 |
| Vitamin A (µg RE) | 207 | 400 | 519 | 222 | 400 | 540 | 238 | 400 | 560 |
| Folic acid (µg) | 48 | 80 | 132 | 50 | 80 | 160 | 133 | 80 | 188 |
| Calcium (mg) | 232 | 280 | 375 | 246 | 280 | 438 | 360 | 280 | 502 |
| Iron (mg) | 9.2 | 6.0 | 9.0 | 9.2 | 6.0 | 10.4 | 5.7 | 6.0 | 11.9 |
| Zinc (mg) | 3.8 | 8.0 | 9.1 | 3.9 | 8.0 | 9.5 | 3.8 | 8.0 | 9.8 |
| Choline (mg) | 47 | 13 | 145 | 55 | 13 | 152 | 114 | 13 | 158 |

*Nutrient requirements from complementary food calculated as total requirement less provided in breastmilk. Estimated mealie meal consumption is 125 kcal, 250 kcal, and 500 kcal based on observed intakes during SHINE trial. IYCF diet includes SHINE IYCF behaviour change modules +20 g Nutributter per day. IYCF-PLUS diet includes IYCF behaviour change modules +20 g Nutributter per day+whole egg powder, moringa powder, and iron and zinc-fortified sugar bean powder.
IYCF, infant and young child feeding; SHINE, Sanitation Hygiene Infant Nutrition Efficacy.

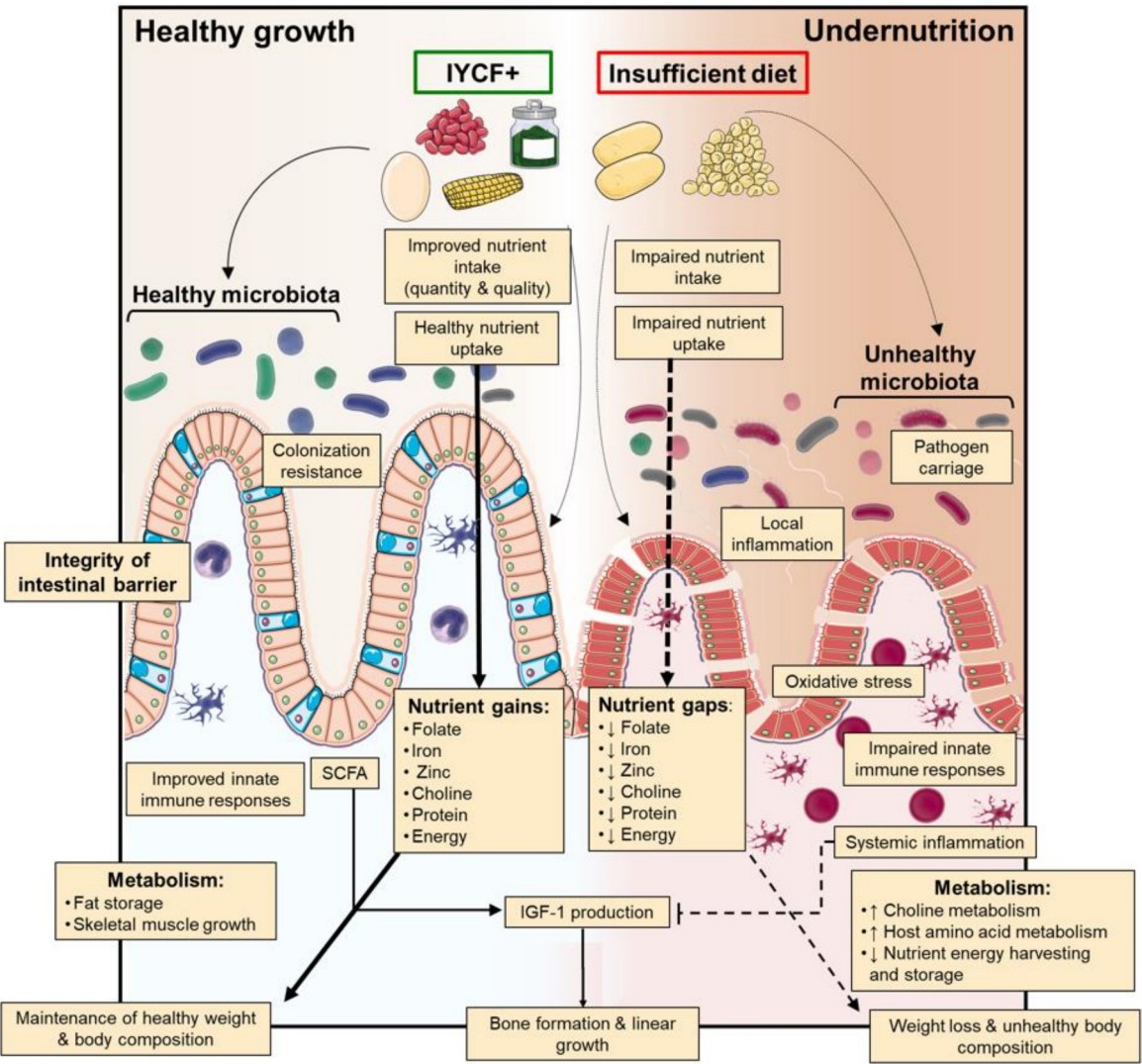

**Figure 1** Hypothesised impact of IYCF-plus intervention on barriers to nutrient intake, uptake and utilisation for healthy child growth. IYCF, infant and young child feeding.

nutrient that promotes growth in animal models.[29] Children with EED have reduced circulating concentrations of phosphatidylcholines,[30] which are required for chondrogenesis at the growth plate, and reliant on adequate dietary intake of choline. One egg meets the majority of an infant's daily requirement. Eggs are also high in fat, energy-dense and make modest but important contributions to vitamin A, iron and zinc intake.

Together, these food supplements have the added plausible benefit of improving the microbiota and gut barrier function and reducing intestinal and systemic inflammation (figure 1). The IYCF-plus intervention will increase dietary diversity and micronutrient content, which has been shown to promote healthy gut microbiota composition.[31] The resistant starch present in legumes is readily fermented by the gut microbiota to produce short-chain fatty acids, which act as a primary energy source for enterocytes, and other metabolites that maintain the integrity of the intestinal barrier.[32] Recent trials suggest that legumes modestly improve linear growth and intestinal

permeability[33 34]; these effects may be enhanced through integration with other micronutrients such as vitamin A and zinc, which can improve gut barrier function.[35 36] Legume intake and high dietary quality scores have been associated with reduced systemic inflammation.[37] Finally, preclinical studies have reported a role for moringa in reducing oxidative stress and improving immune function.[38] We hypothesise that children receiving the IYCF-plus intervention will have a more mature microbiota, reduced EED, less systemic inflammation and an associated improvement in antipathogen immune cell function compared with the IYCF intervention.

Ultimately, through improved agriculture and animal husbandry practices families could become self-sufficient in producing the beans, biofortified maize, moringa and eggs used for the IYCF-plus intervention for their household. As these foods are already grown locally on a small scale, there is also potential for local commercial production and processing of these products that would allow public distribution or purchasing instead of direct

household-level production. Furthermore, these inputs have useful synergies, for example, moringa pods and maize bran provide food for chickens that improves their feed efficiency and increases egg production.

## FORMATIVE RESEARCH AND COMMUNITY ENGAGEMENT

Formative work explored delivery and acceptability of the new food supplements proposed in CHAIN.[39] Briefly, this qualitative study purposively sampled nine CHWs from the Shurugwi rural community where SHINE was conducted, in addition to 27 caregivers of children between 6 and 18 months of age. The aims of this formative work were to assess feasibility of delivering bean, egg and moringa powder to families; the acceptability of recipes devised to incorporate the new supplements into usual foods; and to inform IYCF-plus behavioural modules. The formative work also tested feasibility and uptake of this multicomponent complementary feeding and behaviour change strategy among similar rural households to the study setting.

Activities included focus group discussions with mothers, group-based recipe formulation, standardisation, testing and review as well as home visits to assess ingredient uptake, usage, storage and recipe adherence and innovations. Household observations and views from extended family members indicated high acceptability of the new ingredients. Sensory evaluation by mothers who formulated and standardised the recipes indicated high acceptability of the complementary food recipes. All formative study participants participated in developing the behaviour change messages and finalisation of the recipes in a recipe book developed for use in the CHAIN trial IYCF-plus intervention. The formative research demonstrated feasibility of implementing this multicomponent IYCF intervention comprising behaviour change counselling to promote optimal complementary feeding practices, building self-efficacy of caregivers through cooking demonstrations at home and provision and promoting use of nutrient-dense biofortified maize, moringa, NUA45 and egg powders to feed young children. Uptake of the multicomponent IYCF intervention was high. Household observations and sensory evaluation indicated high acceptability of the new food ingredients and multicomponent IYCF intervention.[39]

## STUDY SETTING AND RECRUITMENT

The CHAIN trial was conducted in Shurugwi district, Zimbabwe. This is a predominantly rural, subsistence farming area, with 15% antenatal HIV prevalence and 35% stunting prevalence.[5] CHWs are a community-level cadre of healthcare workers within the Ministry of Health and Child Care. All CHWs in the study area underwent a 2-week training on the interventions and have monthly supportive supervision meetings with study intervention nurses (INs). Additionally spot checks on performance are conducted. They sensitised community stakeholders

and individual families in their catchment area between birth and 5 months of age about the CHAIN study and refer those who are interested to the trial team. A total of 282 infants turning 5 months of age from April 2021 to August 2021 were identified through CHW registers. Consent visits were scheduled as close as possible to children turning 5 months old, and continued until the required sample size was reached. All children who fulfilled the inclusion and exclusion criteria during enrolment were eligible for the trial, including households who had previously participated in the SHINE trial or the formative research with another child.

A research nurse visited the family's homestead to screen the child for eligibility, provide information on the trial, and undertake written informed consent with the parent/legal guardian in Shona or Ndebele. If the caregiver was not available, visits were rescheduled. All household members were encouraged to be present for consent and subsequent intervention and research visits. Interventions, delivered by CHWs, started as soon as possible after randomisation, and the primary outcome is measured by research nurses at endline. Recruitment of study participants started on 26 April 2021 and study participants were followed until the end of February 2022.

## TRIAL OUTCOMES

The primary outcome is energy intake in kcal at 9 months of age (visit window 9–11 months), as measured by multipass 24-hour dietary recall. Secondary and tertiary outcomes are defined in table 2.

## INCLUSION CRITERIA

Inclusion criteria were assessed by a screening questionnaire delivered to the primary caregiver by a research nurse.

▶ Individual level: Child age between 5 and 6 months.
▶ Household level: Planning to live in the study area for the duration of the trial.

## EXCLUSION CRITERIA

▶ Severe infant disability that interferes with feeding.
▶ Known allergy to peanuts or eggs.

## BASELINE DATA COLLECTION

Baseline data were collected via questionnaire on maternal, infant and household characteristics (table 3). Maternal and infant height (ShorrBoard), weight (Seca 874DR Mother-Baby scale), head circumference and mid-upper arm circumference (ShorrTape) were measured. Maternal HIV status was collected by self-report and review of handheld records. Infant samples of stool, urine and blood were collected for laboratory analyses, including immunology, microbiome and metabolic assays (full details below). Infant blood was collected by venepuncture into heparinised tubes (total 5.4 mL; maximum volume 1 mL/kg), for centrifugation in the

**Table 2** Trial outcomes at 9 months of age (window 9–11 months)

| Endpoint | Definition |
|---|---|
| Energy intake | Percentage of infants meeting daily energy requirements at 9 months of age (window 9–11 months), measured by multipass 24-hour dietary recall. |
| Protein, iron, zinc and folate intake | Percentage of infants meeting daily protein, iron, zinc and folate requirements at 9 months of age, measured by multipass 24-hour dietary recall. |
| Length-for-age Z-score | Length-for-age expressed as a Z-score compared with the WHO 2006 reference median |
| Weight-for-age Z score | Weight-for-age expressed as a Z-score compared with the WHO 2006 reference median |
| Weight-for-length Z score | Weight-for-length expressed as a Z-score compared with the WHO 2006 reference median |
| Haemoglobin | Concentration of haemoglobin (in g/dL) in a whole blood sample, measured by HemoCue point-of-care assay and adjusted for altitude |
| Microbiome maturity | Microbiota-for-age Z-score |
| Environmental enteric dysfunction | Biomarkers of intestinal inflammation (faecal neopterin and myeloperoxidase), small intestinal damage (plasma I-FABP and citrulline), intestinal permeability (faecal A1AT), microbial translocation (plasma sCD14, LBP), systemic inflammation (plasma CRP, AGP, TNF-alpha and K:T ratio) and growth hormone axis (IGF-1) |
| Innate immune cell phenotype | Surface marker expression by peripheral blood monocytes and neutrophils |
| Innate immune cell function | Surface marker expression and cytokine secretion from innate immune cells challenged with lipopolysaccharide in vitro relative to unstimulated controls Capacity of innate immune cells to internalise bacteria in vitro |
| Plasma essential amino acids | Plasma concentrations of phenylalanine, valine, threonine, tryptophan, methionine, leucine, isoleucine, lysine and histidine, as measured by LC-MS-MS |
| Plasma choline | Plasma concentration of choline, as measured by LC-MS-MS |
| Urinary metabolic signature | Global untargeted metabolomic phenotyping undertaken by $^1$H nuclear MR spectroscopy |

AGP, alpha-1 acid glycoprotein; CRP, C reactive protein; LBP, lipopolysaccharide binding protein; LC-MS-MS, liquid chromatography-mass spectrometry.

field laboratory to obtain plasma and peripheral blood cells for storage. One drop of blood was used to measure point-of-care haemoglobin, using a HemoCue 301 machine. Infant urine was collected by applying an adhesive urine bag to the infant's nappy area and waiting for the infant to pass urine during the visit. Urine was poured from the bag into a plain storage tube for transport to the field laboratory in a cool box. Infant stool was collected from the nappy into a plain tube and stored in a cool box for transport to the laboratory. If the infant did not pass stool during the visit, the mother was provided with a collection pack and instructions for how to collect the specimen the next morning, or as soon as possible thereafter, and keep the sample in a cool part of the house. The mother was asked to contact the study team once the sample has been collected and the research nurse visited the home to collect the sample. The research nurse checked the sample on arrival, labelled it with a barcode and placed it into a cool box for transport to the field laboratory. Children with symptomatic mild to moderate anaemia (<11 g/dL) or with severe anaemia (<7 g/dL) were referred to local clinics. Children with moderate or severe acute malnutrition (MUAC <125 mm or weight-for-length Z-score <−2) were also referred to local clinics.

## RANDOMISATION

The randomisation schema was prepreparad by the trial statistician using the RALLOC command in STATA V.14, using random permuted blocks of varying block sizes, with a 1:1 allocation to IYCF or IYCF-plus. Randomisation codes are securely embedded in the trial database so that the next number is accessible to the data officer, but not the entire randomisation list. Participant IDs were pregenerated and allocated to treatment arms prior to recruitment into the study. Participant IDs were assigned to a specific participant within a household after consent. Twins or eligible infants within the same household were allocated to the same trial arm. One of each twin were randomised to a specific arm. The CHW, supervised by an IN, visited the mother to tell her the trial allocation and to begin the interventions. It was not possible to blind households or fieldworkers to the interventions, but data and laboratory analysts are blinded to the allocated arm. All laboratory and data analyses will be identified by participant ID number, which does not contain details of the trial arm, and then merged by the trial statistician before reporting. Monthly reports of adverse events were reported to an independent trial safety monitor and to the Medical Research Council of Zimbabwe.

**Table 3** Chain trial schedule*

| Procedure | Visit 1†<br>Screening and<br>enrolment | | Visit 2<br>Baseline | Visit 3c<br>Endline |
|---|---|---|---|---|
| Screening and eligibility check form 2 | x | Randomisation and start of monthly IYCF or IYCF-plus interventions | | |
| Informed consent<br>form 3 and 4 | x | | | |
| Locator and contact information<br>form 5 | x | | x | x |
| Baseline interview<br>form 6 | | | x | |
| Maternal weight, height and MUAC | | | x | |
| Infant weight, height, MUAC and head circumference | | | x | x |
| Infant blood collection§ | | | x | x |
| Infant haemoglobin | | | x | x |
| Infant stool collection‡ | | | x | x |
| Infant urine collection§ | | | x | x |
| Endline interview<br>form 9 | | | | x |
| 24-hour dietary recall¶<br>form 10 | | | | x |
| Informed consent for qualitative substudies**<br>form 11 | | | | |
| Qualitative substudy 1 guide form 12a | | | | x |
| Informed consent for qualitative sub studies<br>Form 11†† | | | | |
| Qualitative substudy 2 guide<br>form 12b | | | | x |

*Trial enrolment began in April 2021 and concluded in August 2021.
†For mothers who wish to provide informed consent on a subsequent day to screening, these visits will be separated.
‡Target date 9 months of infant age (visit window 9–11 months or 274–334 days).
§If any specimens cannot be collected during the visit (eg, if the infant fails to pass stool), the specimen collection will be rescheduled for the next day, or as soon as possible after the visit. Rarely it may be necessary to repeat a specimen collection if the sample was insufficient, or fails quality control checks during processing in the laboratory. All blood draws will be kept to a safe limit, defined as maximum of 1 mL/kg body weight.
¶The primary outcome, measured by multipass 24-hour dietary recall, will be repeated 1 week later in a subsample of 50% infants for methods validation.
**Subgroup of up to 20 purposively selected households. The social scientist will visit the family during a separate visit at 7–9 months (214–273 days) of infant age, and obtain separate written informed consent for the qualitative interviews.
††Subgroup of up to 30 purposively selected households. The social scientist will obtain written informed consent for the qualitative interviews regarding migration, which will be conducted throughout the course of the study.
IYCF, infant and young child feeding; MUAC, mid-upper arm circumference.

## INTERVENTION DELIVERY

Behavioural modules: A total of 9 interpersonal face-to-face counselling modules were delivered to caregivers in each arm by CHWs during 10 home visits, which coincide with key infant ages, so that sequential age-appropriate messages about complementary feeding are introduced and reinforced (table 4). The CHW introduces the food supplements in both arms, demonstrates how to add them to food, monitors for any adverse reactions and provides monthly resupplies. Modules are interactive and are delivered to all household members present. The last module was delivered at 11 months of age, with infant food supplements provided until 12 months of age, when all trial interventions end. Using this design, we ensured that all infants are still receiving the IYCF or IYCF-plus interventions when endline data collection occurred at 9 months (window 9–11 months) of age.

If a module was missed, the CHW attempted to catch up by scheduling a new date, or else summarised the missed module at the next scheduled visit. If a caregiver moved within the study area, the CHW covering that area delivered modules to the caregiver, where possible; if the caregiver moved out of the study area, she did not receive study modules or food supplements. Endline data collection visits are conducted regardless of where the caregiver moves to. The estimated length of time exposed to the

**Table 4** Module delivery

| Infant age (window period)* | Home visit | Infant age (window period)* | Control arm IYCF modules and supplies | Intervention arm IYCF-plus modules and supplies |
|---|---|---|---|---|
| 5 months (5 months to <6 months) | 1 | 5 months (5 months- <6 months) (153–182 days) | Module 1<br>▶ Nutrition for your baby<br>▶ Composition and functions of SQ-LNS (Feeding as a chore or actual work; stomach capacity and graph on nutrient gap of breastmilk and mealie meal) | Module 1<br>▶ Nutrition for your baby+introduction of the three powders<br>▶ Composition and functions of SQ-LNS and the three powders (Feeding as chore or actual work; stomach capacity and graph on nutrient gap of breastmilk and mealie meal) |
| 6 months (6 months to <7 months) | 2 | 6 months (6 months to <7 months) (183–213 days) | Module 2<br>▶ Introducing solid foods<br>▶ Additional information on SQ-LNS<br>▶ Breastmilk+porridge + SQ-LNS | Module 2<br>▶ Introducing solid foods<br>▶ Additional information on SQ-LNS and the three foods (Bean powder, Egg powder and Moringa powder)<br>▶ Breastmilk+porridge + SQ-LNS |
| 6 months 1 week (6 months 1 week to <7 months) | 3 | 6 months 1 week (6 months 1 week to <7 months) (190–213 days) | Module 2.1<br>▶ Frequency of complementary foods<br>▶ Reinforce messages<br>▶ Breastmilk+porridge+SQ-LNS | Module 2.1<br>▶ Introduce Bean Powder<br>▶ Frequency of complementary foods<br>▶ Reinforce messages<br>▶ Breastmilk+porridge + SQ-LNS+bean powder |
| 6 months 2 weeks (6 months 2 weeks to <7 months) | 4 | 6 months 2 weeks (6 months 2 weeks- <7 months) (197–213 days) | Module 2.2<br>▶ Nutrient-dense complementary meals<br>▶ Reinforce messages<br>▶ Breastmilk+porridge+SQ-LNS+nutrient density | Module 2.2<br>▶ Introduce Egg Powder<br>▶ Nutrient-dense complementary meals<br>▶ Reinforce messages<br>▶ Breastmilk+porridge+ SQ-LNS+bean powder+egg powder |
| 6 months 3 weeks (6 months 3 weeks to <7 months) | 5 | 6 months 3 weeks (6 months 3 weeks to <7 months) (204–213 days) | Module 2.3<br>▶ Complementary feeding schedule and family support<br>▶ Reinforce messages<br>▶ Breastmilk+porridge+SQ-LNS | Module 2.3<br>▶ Introduce Moringa<br>▶ Complementary feeding schedule and family support<br>▶ Reinforce messages<br>▶ Breastmilk+porridge + SQ-LNS+bean powder+egg powder+moringa powder |
| 7 months (7 months to <8 months) | 6 | 7 months (7 months to <8 months) (214–243 days) | Module 3<br>▶ Introducing more foods<br>▶ Reinforce messages<br>▶ Breastmilk+SQ LNS+porridge (sadza/ soup/potatoes/maheu+other locally available foods, eg, vegetables) | Module 3<br>▶ Introducing more foods<br>▶ Reinforce messages<br>▶ Breastmilk+SQ LNS + bean powder+egg powder+moringa powder+porridge (sadza /soup/ potatoes/maheu+other locally available foods, eg, vegetables) |
| 8 months (8 months to <9 months) | 7 | 8 months (8 months to <9 months) (244–273 days) | Module 4<br>▶ Feeding during illness<br>▶ Reinforce messages<br>▶ Breastmilk+SQ LNS+porridge (sadza/ soup/potatoes/maheu+other locally available foods, eg, vegetables) | Module 4<br>▶ Feeding during illness<br>▶ Reinforce messages<br>▶ Breastmilk+SQ LNS + bean powder+egg powder+moringa powder+porridge (sadza /soup/ potatoes/maheu+other locally available foods, eg, vegetables) |

Continued

**Table 4** Continued

| Infant age (window period)* | Home visit | Infant age (window period)* | Control arm IYCF modules and supplies | Intervention arm IYCF-plus modules and supplies |
|---|---|---|---|---|
| 9 months (9 months to <10 months) | 8 | 9 months (9 months to <10 months) (274–304 days) | Module 5 <br> ▶ Dietary diversity <br> ▶ Reinforce messages <br> ▶ Dietary diversity <br> ▶ Breastmilk+SQLNS + porridge (sadza/soup/potatoes/maheu+other locally available foods, eg, vegetablestc) | Module 5 <br> ▶ Dietary diversity <br> ▶ Reinforce messages <br> ▶ Dietary diversity <br> ▶ Breastmilk+SQLNS + bean powder+egg powder+moringa powder+porridge (sadza /soup/potatoes/maheu+other locally available foods, eg, vegetables) |
| 10 months (10 months to <11 months) | 9 | 10 months (10 months to <11 months) (305–334 days) | Module 6 <br> ▶ Reinforcing all Modules <br> ▶ Breastmilk+SQLNS+porridge (sadza/soup/potatoes/maheu+other locally available foods eg, vegetables) | Module 6 <br> ▶ Reinforcing all Modules <br> ▶ Breastmilk+SQLNS + bean powder+egg powder+moringa powder+porridge (sadza /soup/potatoes/maheu+other locally available foods, eg, vegetables) |
| 11 months (11 months to <12 months) | 10 | 11 months (11 months to <12 months) (345–365 days) | Module 6 <br> ▶ Reinforcing all Modules <br> ▶ Breastmilk+SQLNS+porridge (sadza/soup/potatoes/maheu+other locally available foods, eg, vegetables) | Module 6 <br> ▶ Reinforcing all Modules <br> ▶ Breastmilk+SQLNS + bean powder+egg powder+moringa powder+porridge (sadza /soup/potatoes/maheu+other locally available foods, eg, vegetables) |

*Each module session will be delivered for approximately 60 min. If a module is not delivered within the intervention window (ie, appropriate infant age), the CHW will try to catch up by scheduling a new date as soon as possible. Each module will be delivered to the mother and her family. If the rescheduled module for modules 1.0 and 2.0 overlap, these two modules will be delivered at the same time. If the rescheduled module overlaps with the next visit for other modules (2.0, 2.1, 2.2, 2.3, etc) the visits will be scheduled at least 3 days apart so that families have time to absorb the new material. Delivery of IYCF-plus modules has therefore been designed to be flexible following complementary feeding guidance. Experience from formative work showed that it is feasible to deliver the combined modules at once.
CHW, community health worker; IYCF, infant and young child feeding; SQLNS, small-quantity lipid-based nutrient supplements.

intervention is between eight and eleven months, from 5 to 15 months. Median estimated length of time exposed to the intervention is 8 months.

### IYCF arm

Core IYCF counselling modules were complemented by provision of one sachet of 20 g SQ-LNS daily between 6 and 12 months of age. SQ-LNS is a peanut-based supplement rich in calories, protein and micronutrients, which can be consumed directly from the sachet or mixed with porridge. Families also received a daily infant ration of white maize to feed the baby as porridge.

### IYCF-plus arm

As in the IYCF arm, core counselling modules focusing on complementary feeding were delivered by the CHW, with provision of one sachet per day of 20 g SQ-LNS between 6 and 12 months of infant age. In addition, families received NUA-45 biofortified bean powder, whole egg powder and moringa leaf powder for provision to the study child. The quantity of food supplements provided was based on the child's age to ensure the daily recommended nutrient intake (RNI) was met if the food supplement was consumed (table 5). Families also received a daily infant ration of PVA biofortified maize to feed the baby as porridge. Supplements were delivered in sealed containers, which the mother is asked to keep in a cool

part of the house. Six recipes promoting high-quality staple foods, which were developed and standardised in the formative studies,[39] are outlined in a recipe book, with cooking demonstrations given by the CHW.

### INTERVENTION DELIVERY AND UPTAKE

Each CHW delivered the intervention to 1–3 enrolled households during the study. Eight intervention nurses (INs) (separate from the research nurses) were responsible for monitoring delivery of modules and food supplements and evaluating intervention uptake and compliance to recommended behaviours by caregivers. INs did not provide counselling to mothers but did provide supportive supervision to CHWs by scheduled attendance at some household visits to provide feedback and by conducting unscheduled spot checks. An Intervention nurse sits in each time a CHW was delivering a module for the first time and additionally if needed. INs held monthly meetings with the CHWs they supervise, to share learning, capture data on module delivery and provide retraining as needed. A module delivery and intervention uptake checklist was completed by CHWs at each module delivery visit, and submitted to INs during monthly meetings. The checklist recorded modules that have been successfully delivered and dates of delivery.

**Table 5** Quantities of food supplements in the IYCF-plus arm

| Infant age group (months) | Mealie meal (PVA maize)* | SQ-LNS† | Whole egg powder‡ | Moringa leaf, dried and ground§ | Sugar bean legume, finely ground¶ |
|---|---|---|---|---|---|
| 6–8 | ≥42 g (3 Tbsp) | 20 g | 14 g (3 tsp) | 5 g (1 tsp) | 5 g (1 tsp) |
| 9–11 | ≥71 g (4.5 Tbsp) | 20 g | 14 g (3 tsp) | 10 g (2 tsp) | 10 g (2 tsp) |

Food supplements will be delivered monthly by CHWs.
Tbsp - Tablespoon
tsp - Teaspoon
*Mealie Meal (PVA maize) will be provided in 500 g bags, with 3 bags/month (1500 g) between 6 and 8 months of age and 5 bags/month (2500 g) between 9 and 11 months of age. Households in the IYCF arm will receive the same amount of mealie meal per month
†SQ-LNS will be supplied monthly to ensure 1×20 g sachet per day can be provided (30 or 31 sachets per month).
‡Whole egg powder will be delivered as a 500 g bag per month.
§Moringa leaf powder will be supplied in 175 g bags, with 1 bag/month (175 g) between 6 and 8 months of age and 2 bags/month (350 g) between 9 and 11 months of age.
¶NUA 45 sugar bean powder will be supplied in 175 g bags, with 1 bag/month (175 g) between 6 and 8 months of age and 2 bags/month (375 g) between 9 and 11 months of age. These quantities allow for 15% extra in case of spillage or sharing.
CHW, community health worker; IYCF, infant and young child feeding; PVA, pro-vitamin A; SQ-LNS, small-quantity lipid-based nutrient supplement.

Data on uptake of interventions and compliance to recommended behaviours will include utilisation of food supplements, any sharing of food supplements observed and involvement of other family members in child feeding assessed by a caregiver questionnaire.

## FOLLOW-UP DATA COLLECTION

Households were visited by a research nurse at 9 months of infant age (window 9–11 months) for endline data collection. If the child was not present, the visit was rescheduled. Infant weight, length, MUAC and head circumference were measured, and samples of blood, urine and stool collected, using the same methods as at baseline. Children with illness were referred to local clinics, using the same criteria as at baseline.

The trial primary outcome was measured by research nurses via 24-hour multipass dietary recall. A subgroup of 50% of randomly selected household had a second 24-hour dietary recall visit approximately 1 week later. This method provides a robust and validated measure of nutrient intake based on a comprehensive and standardised assessment.[40] The dietary recall method assesses all food and beverages consumed in the previous 24 hours (including supplements provided by the trial) and comprises five passes. In the first pass, the research nurse asks the caregiver to list all foods consumed by the child during the previous day, and to list any night feeds. In the second pass, the caregiver is asked to list all activities they undertook, and whether they fed the child food between activities; this helps the caregiver remember all feeding episodes. In the third pass, more details about foods and beverages are collected, including the time and place of preparation, ingredients and brand of foods given. In the fourth pass, the caregiver estimates the portion size fed to the child. Research nurses carry samples of the most consumed foods and ask the caregiver to estimate the amount fed to the child. The fieldworker then transfers the estimated portion to a standard cup, spoon or digital scale for recording. In the final pass, the caregiver recalls if there were any foods or meals that have not already been mentioned. Caregivers are also asked about the general health of the child on the previous day, whether the child's intake was less or more than usual, and how many times the child was breastfed.

Data from the 24-hour recall will be converted to observed energy and nutrient intakes by the following steps:

1. Ingredients and portion sizes were measured and weighed in grams where possible. For ingredients that could not be weighed, they will be converted to grams using locally collected data on food densities, supplemented with food density data from the Food and Agriculture Organisation, United State Department of Agricultire and Nutrient Database for Standard Reference.[41–43]
2. Mixed dishes were disaggregated into ingredients and entered into NutriSurvey to calculate nutrients in 100 g of food.
3. Energy from each individual food/ingredient will be estimated using food composition data from regional food databases and USDA databases that have been collated for use in Zimbabwe over several studies.[44]

Estimated energy and nutrient intakes will be compared with WHO-estimated energy requirements, the Institute of Medicine, Food and Nutrition Board recommended daily allowances for protein and choline, and the WHO-RNIs for other vitamins and minerals. For breastfeeding children, we will calculate the required nutrient intake from complementary foods by subtracting the amount of each nutrient in 550 g breast milk from the total requirement which is the estimated intake of breast milk for children aged 9–11 months old.[45] Energy requirements are calculated as kilojoules required per kilogram of body weight for breastfed children[46]; however, we will apply the slightly higher energy requirement estimated by Butte for children in low-income settings, which reflect increased needs due to greater infection burden.[47] Protein requirements will be defined as the WHO-recommended safe level of protein intake for children aged 9–11 months old.[48] Fat requirement will be defined as 35% energy requirement, which is the midpoint of several recommendations.[49] Micronutrient requirements will be defined as WHO-RNIs, except for calcium which is

defined as the mean of the WHO RNI and US RNI. For zinc and iron, we will assume 30% and 10% bioavailability, respectively.[29 46 50 51] For breastfed children, we will estimate the required nutrient intake from complementary foods by subtracting the amount of each nutrient in breast milk from the total requirement.[51] Using this comprehensive approach, we will determine the impact of the IYCF-plus vs IYCF intervention on energy intake (primary outcome) and the relative contributions of supplements (including SQ-LNS) and other complementary foods in closing infant nutrient gaps across trial arms. In addition to assessing total protein intake, we will explore essential amino acid intake, digestibility-adjusted protein intake and inflammation-adjusted protein intake.[52]

## TRAINING OF CHWS, INTERVENTION AND RESEARCH NURSES

Eight INs underwent a 2-week training on delivery of the trial interventions and provide supportive supervision to CHWs. All CHWs in the study area underwent a 2-week training on the interventions. A training cascading approach was used where INs took part in training of CHWs, supporting other research staff. Monthly supportive-supervision cluster meetings with CHWs were conducted by INs. Four research nurses (Data Collectors- DCs) underwent a 4-week training on conducting consenting, baseline, endline and 24-hour dietary recall interviews. Research nurses were trained separately from INs and all activities they carried out were conducted separately to avoid research bias.

## DATA COLLECTION AND MANAGEMENT

Research data were collected onto electronic case report forms (eCRFs) using preprogrammed tablets, with Open Data Kit software. Full data validation procedures were programmed into the tablets including embedded skip patterns, data completeness and plausibility checks. All data were checked daily by the field data officer, with implausible values verified or recollected. Back-up paper CRFs are carried by research nurses in the event of tablet failure. Data are uploaded from tablets onto a secure trial database daily and backed up onto a secure cloud database hosted on Microsoft Azure. Data will be stored for 20 years.

Each participant is allocated a participant identifier which is used on all forms to identify the child. Personal information and data are kept confidential and managed in accordance with the requirements of the Medical Research Council of Zimbabwe. Paper records (eg, CRF, clinical/laboratory information and test results) will be entered into the electronic database; source documents will be stored in a secure, locked cupboard at each study site, and kept fully confidential. Data will be kept securely on a password-protected customised MS-SQL Server trial database and hosted by Microsoft Azure.

## SAMPLE MANAGEMENT

Preprinted barcodes identifying the participant ID and sample type were adhered to collection tubes in the field, which were transported to the laboratory at room temperature (for blood) or in a cooler bag (for stool and urine samples). When samples arrived at the laboratory, they were processed and aliquoted into cryovials which were labelled with barcodes identifying the participant ID, sample type and aliquot number. Samples were stored in the field laboratory at −80°C. At regular intervals, samples were transported to the main laboratory in Harare, where they were stored at −80°C until analysis or shipment. All samples will be shipped to external laboratories on dry ice. Sample lists will be maintained in the main trial database. If participants consented to long-term storage, samples will be stored for up to 20 years.

## SAMPLE SIZE

The sample size of 192 infants assumes 10% lost to follow-up due to withdrawal and infant deaths, meaning there will be an estimated 86 evaluable infants per group at endline. This sample size provides 86% power at 5% significance to detect a 20% increase in the proportion of infants achieving their recommended energy intake (by 24-hour dietary recall) in the IYCF-plus arm, assuming that only 65% of infants are meeting requirements in the IYCF (standard-of-care) arm based on SHINE data. If lost to follow-up is as high as 15%, we will still have 80% power to detect a 20% increase in the proportion reaching their daily energy intake (study primary outcome).

## STATISTICAL ANALYSIS

Analysis of trial outcomes will be by intention to treat. P values will be two sided and interpreted as significant if $p<0.05$. Binary outcomes will be compared between groups using the $\chi^2$ test and logistic regression to compute ORs and corresponding 95% CIs. Other categorical outcomes with more than two levels will be compared between groups using $\chi^2$ tests and multinomial regression. Continuous outcomes will be compared using simple t-tests and linear regression. Non-normal continuous outcomes will be transformed appropriately before analysis. Robust SE estimates will be used to estimate CIs.

The primary outcome, percentage of infants meeting energy intake (kcal/day), will be calculated using measured intake data from dietary recalls from all participants. Dietary energy intake (DEI) was collected through an interactive, multiple-pass, 24-hour recall at the 9-month endpoint among all children, and in 50% of children through an additional dietary recall 7 days later. The DEI estimate will be used with a logistic regression model clustering for the repeat measure using the multilevel mixed-effects logistic regression method. This method will be implemented within STATA. It is well recognised that the dietary intake data contain a lot of uncertainty, and thus, we will conduct a sensitivity analysis using the National Cancer Institute (NCI) method calculated for usual overall average intake.

We will estimate the mean and percentiles of usual energy intake distributions using the NCI method.[53 54] This method adjusts for measurement error—primarily due to day-to-day variability in intakes—in observed, single-day estimates of nutrient intakes. We will use the NCI macros, DISTRIB and MIXTRAN, plus bootstrapped SEs for hypothesis testing.[25 27] The MIXTRAN macro fits a mixed effects model of usual energy intake, and the DISTRIB macro uses a Monte Carlo procedure to estimate percentiles of the usual intake distribution. We will use the MIXTRAN macro to fit a model of energy intake with a fixed effect for intervention group and the DISTRIB macro to estimate usual energy intake distributions by treatment group. We will bootstrap the parameters to estimate the SE of the mean usual intakes. We will use the point estimates and SE estimates to construct 95% CIs and calculate p values for difference in mean by intervention group based on Welch's t-test. Bootstrapped SEs will be calculated using bootstrap samples. When intake or the number of repeat recalls is low, it is common for the NCI method to fail to converge. In cases where convergence prohibits bootstrapping, we will test hypotheses using the Wilcoxon testing for clustered method.

We will also interpret the mean intake of each nutrient for the primary and secondary analysis in each arm and will present mean differences with the 95% CI for interpretation.

All analyses will be prespecified in a statistical analysis plan and posted online at Open Science Framework before analyses begin (https://osf.io/njy2a/). A per-protocol analysis will be conducted using adherence data to predefine the per protocol population. Adherence will be assessed by self-report in a 7-day recall. We will undertake two subgroup analyses: (1) by infant sex and (2) by maternal HIV status, if we find some evidence of interactions with the intervention (p<0.10).

## LABORATORY ANALYSES

Biological samples will be used to evaluate the nutrient profiles, EED, systemic inflammation, innate immune function, metabolic phenotype and the gut microbiota, as shown in table 6. This is a tertiary outcome being measured on everyone with samples available for baseline and endline. The study is not powered for this outcome and is exploratory.

### EED and systemic inflammation

We will use a combination of plasma and stool ELISA assays to compare the impact of IYCF and IYCF-plus on EED, by characterising the hypothesised causal pathway from the gut to growth, measuring markers of intestinal inflammation (stool myeloperoxidase, neopterin) small intestinal damage (plasma I-FABP), intestinal permeability (alpha-1 antitrypsin), microbial translocation from the gut (plasma LBP, sCD14), systemic inflammation (AGP, C reactive protein, TNFα) and growth hormone activity (plasma IGF-1). We will also measure aflatoxin M1 in urine, to assess recent exposure to dietary aflatoxin, which is a plausible cause of EED

**Table 6** Laboratory analyses

| Sample type | Assay | Method | Location of work | Study subjects | Time-points |
|---|---|---|---|---|---|
| Plasma | I-FABP, CRP, AGP, TNFα, LBP, sCD14, IGF-1 | ELISA | Zvitambo | All | Baseline, endline |
| Peripheral blood leucocytes | Innate immune cell phenotype* | Flow cytometry | Zvitambo | All | Baseline, endline |
| Whole blood | Whole Blood Culture with and without LPS | Cell culture, flow cytometry and ELISA | Zvitambo | All | Baseline, endline |
| Whole blood | Bacterial binding Assay | Cell culture, flow cytometry | Zvitambo | All | Baseline, endline |
| Stool | Myeloperoxidase, neopterin, alpha-1 antitrypsin | ELISA | Zvitambo | All | Baseline, endline |
| Urine | Global untargeted metabolomic phenotyping | $^1$H NMR spectroscopy | Southampton, UK | All | Baseline, endline |
| Plasma | Global untargeted metabolomic phenotyping | $^1$H NMR spectroscopy | Southampton, UK | All | Baseline, endline |
| Plasma | Kynurenine:tryptophan ratio, citrulline, essential amino acids, choline | Ultrahigh-performance liquid chromatography tandem mass spectrometry with electrospray ionisation | Southampton, UK | All | Baseline, endline |
| Stool | Whole metagenome shotgun sequencing | Illumina HiSeq | Blizard Institute, UK | All | Baseline, endline |
| Stool | Metabolic phenotyping of faecal water | $^1$H NMR spectroscopy | Southampton, UK | All | Baseline, endline |
| Urine | Aflatoxin M1 and creatinine | ELISA | Zvitambo | All | Baseline, endline |

*Expression of surface activation markers HLA-DR, CD64 and CD16 on monocytes and HLA-DR, CD64, CD16 and CD62L on neutrophils.
AGP, alpha-1 acid glycoprotein; CRP, C reactive protein; I-FABP, intestinal fatty acid binding protein; IGF-1, insulin like growth factor 1; LBP, lipopolysaccharide binding protein; LPS, lipopolysaccharide; NMR, nuclear MR.

and may be reduced in the IYCF-plus arm since PVA maize appears less prone to fungal contamination.

### Immune function

Immune cell activation is metabolically costly and chronic activation by recurrent infections/EED may create a barrier to children meeting their nutrient requirements by: (1) driving inflammation and oxidative stress, (2) contributing to enteropathy and (3) compromising the capacity of innate immune cells to defend against new infections, which further deplete dietary nutrients. Multiple innate and adaptive immune mediators are dysregulated in undernourished children,[55] but little is known about if/how nutritional interventions affect immune defences.[56] We will use whole blood samples to compare innate immune cell phenotype and function between randomised groups. We will quantify surface expression of activation markers HLA-DR, CD64 and CD16 on blood monocytes, and CD64 and CD62L on blood monocytes and neutrophils via flow cytometry. To characterise the functional capacity of innate immune cells to respond to pathogen challenge, we will quantify proinflammatory cytokine secretion in supernatants derived from whole blood cultures with and without bacterial lipopolysaccharide. Whole blood culture with and without fluorescent-labelled *Escherichia coli*-coated bioparticles will be used to quantify uptake of bacteria via flow cytometry.

### Metabolic phenotyping

A targeted ultra-performance liquid chromatography-mass spectrometry-based assay will be used to measure tryptophan-related metabolites in plasma (funding permitting).[57] This includes metabolites involved in the kynurenine, serotonin and indole pathways. In addition, downstream NAD+ related metabolites, such as nicotinic acid, nicotinamide and nicotinamide riboside will be measured together with markers of systemic inflammation (neopterin), enterocyte mass (citrulline) and the neurotransmitter dopamine. $^1$H nuclear MR spectroscopy will be used to characterise the metabolic profiles of urine, plasma and faecal water samples. This approach measures H-containing metabolites present above the limit of detection in the samples in an untargeted manner. This captures information on amino acids, gut microbial metabolites and metabolites involved in choline and energy metabolism. It may also be used to study dietary components and assess variation in their digestion.

### Microbiome sequencing

Whole metagenome shotgun sequencing will be employed on stool samples to examine the effect of the trial interventions on the gut microbiome and its association with growth. DNA will be extracted from stool aliquots (200 mg) using the Qiagen PowerFecal Pro DNA kit, followed by metagenomic sequencing library preparation. Following qualitative and quantitative assessment of sequencing libraries, sequencing will be performed via the HiSeq 2500 platform producing 6–10 million sequencing reads per sample. Following quality control and trimming of human reads, sequencing reads will be processed through validated pipelines to generate compositional (MetaPhlAn V.3) and functional (HUMAnN V.3) readouts of the gut microbiome. Microbiome maturity will be assessed as previously described using a control dataset generated from the SHINE trial.[58] Aliquots of stool stored in glycerol will be used to isolate microorganisms of interest for downstream experiments assessing the influence of the gut microbiome on EED and growth.

## QUALITATIVE STUDIES

The first qualitative substudy will develop a more in-depth understanding of how the CHAIN interventions were integrated into household nutrition practices, and the cultural, economic and social processes that shape this. Up to 20 families were purposively sampled based on type of household, caregiver characteristics and trial arm. Semistructured interviews were conducted to explore a range of themes, including any changes in the participants' receptiveness to the food supplements or children's response to them, the ability to maintain compliance with the food preparation guidelines, and challenges they may have experienced (eg, accessing or storing the food supplements). The interviews also focus on relevant cultural, economic and social processes operating at the household level that may influence food preparation and consumption practices and explore how these shape the sustainability of the intervention. Additionally, emphasis was placed on exploring the role that household and area-level gender dynamics may play in decision-making practices.

A second qualitative substudy will identify the ways in which household migration (defined here as people who have moved away from a household for 3 months or more) influences food security. Thirty participating households were identified from the baseline survey as having at least one family member who has migrated across a range of geographical scales (local, regional, national and international) and where possible representing different household types (including, male-headed or female-headed households, orphan-headed households, elder-headed households). In-depth interviews were conducted with study participants to explore the interactions that exist between household migration and remittance practices (receiving and sending) and their potential to influence household food consumption and production practices. In addition, the in-depth interviews aim to consider the importance of geographical scale to remitting practices and their influence on food consumption and production and investigate possible interactions between migration and household participation in the CHAIN interventions.

Interviews and focus groups were audio recorded, for subsequent transcription and translation; transcripts will be entered into NVivo for coding, analysis and interpretation. Analysis will proceed using both deductive and inductive approaches and will use the framework method often employed in multidisciplinary health-related research. All audio recordings will be destroyed, although transcriptions will be stored securely and made available for future analysis as required.

## TRIAL RISKS AND ADVERSE EVENTS

All interventions are commercially available foods (beans, egg powder, moringa, maize) or food supplements that are widely used globally (SQ-LNS). Peanuts and eggs are both staple foods in rural Zimbabwe; although both are potentially allergenic among infants in high-income settings, the prevalence of food allergies in sub-Saharan Africa is extremely low.[5 59 60] SQ-LNS, which contains peanuts, was used in the same community in the SHINE trial among more than 2000 infants. From over 365 000 doses of SQ-LNS that were given to infants in SHINE, there were no allergic reactions and no serious adverse events.[5] Only two adverse events were possibly or probably related to SQ-LNS; both resolved without sequelae.[5] Children with a known allergy to peanuts or eggs will be excluded from the study to further minimise the risk of adverse events. All adverse events will be reported and reviewed by the study physician, and a tabulated monthly summary will be sent to an independent safety monitor. All serious adverse events and trial-related adverse events will be reported to the Medical Research Council of Zimbabwe according to established time frames.

The COVID-19 pandemic poses a potential risk to research activities in the community. We ensured that research staff wear personal protective equipment and practice physical distancing to keep themselves and research participants safe. All staff were trained in COVID-19 protocols. Interviews were conducted in a confidential outdoor part of the homestead wherever possible. CHWs conduct their activities with guidance for safe working from the Ministry of Health and Child Care. Procedures were reviewed as the pandemic progresses, and any changes discussed with the District Health Executive and with the Medical Research Council of Zimbabwe. The food supplements are stopped at 12–15 months and not continued after the trial.

## ETHICS

Ethical approval was granted by the Medical Research Council of Zimbabwe (MRCZ/A/2679), and Sponsorship provided by Queen Mary University of London (Joint Management Research Office, http://www.jrmo.org.uk/). All caregivers/legal guardians provided written informed consent on behalf of their child. Information and consent procedures were administered in each individual's language of choice (Shona, Ndebele and/or English). Because mothers often need to consult with other family members before deciding about their child's participation in a trial, we include other family members in consent discussions if the mother wishes. If the caregiver wanted to think about participation or consult, the consenting visit was rescheduled to give her time. Illiterate mothers who understand a verbal explanation of the study can provide a thumb imprint on the consent form in the presence of an independent witness. Mothers aged between 15 and 18 years are considered emancipated minors under Zimbabwean law and could consent on behalf of their child. Families were provided with a small gift (soap and Vaseline) for each research data collection visit. Consent forms are available online (https://osf.io/njy2a/).

Since the IYCF intervention in the SHINE trial led to 20% reduction in stunting,[5] we are providing IYCF as the enhanced standard-of-care for the control arm, since we believe it is unethical not to provide this intervention in the exact same population. Provision of food supplements may confer benefits for infant nutrition and growth, and for household well-being, but we believe it is ethically justifiable to randomise the supplements of powdered egg, NUA45 sugar beans, moringa and PVA maize, since none are routinely provided to households in the community and there is scientific equipoise as to whether these food supplements will bring additional benefit to what is already being provided in the IYCF arm.

## DISSEMINATION

Trial results will be presented at international conferences and published in open-access journals. Data will be available on request, after publication of the primary trial findings, by contacting the Trial Management Group, with details of data access requests available on the Zvitambo website. Results will be presented to the Ministry of Health and Child Care in Zimbabwe and will be disseminated to the study district through the Community Engagement Advisory Board which comprises peers selected by the community to review ongoing research studies in the community. Results will also be presented to UNICEF to inform the design and scale-up of IYCF programmes in Zimbabwe.

## PATIENT AND PUBLIC INVOLVEMENT

Research questions and outcome measures were informed from gaps identified in the SHINE trial, incorporating the priorities, experiences and preferences of community members. Community members who participated in the SHINE study, living in the same setting the CHAIN trial was conducted informed CHAIN's research questions, design and test foods used based on their preferences and priorities. Families who participated in the formative study informed the trial design and the burden of the intervention. Trial findings will be disseminated to participants through community engagement meetings coordinated by the Shurugwi Community Engagement Advisory Board.

## DISCUSSION

Stunting remains a global health challenge which hinders human capital and perpetuates poverty. There is an urgent need for more efficacious nutrition-specific interventions to enhance child linear growth during complementary feeding. Currently, multiple barriers constrain nutrient intake, uptake and utilisation, including marginal diets, EED, perturbations of the microbiome, metabolic dysregulation and chronic inflammation. Current IYCF approaches partly close nutrient gaps but require optimisation to fully restore healthy child growth. Using locally available foods

with functional properties to supplement current IYCF approaches could have the dual goal of closing nutrient gaps in infancy and ameliorating pathogenic barriers to nutrient uptake and utilisation. We will test these ideas using a rigorous proof-of-concept trial design in a rural, subsistence farming community with a high burden of stunting, and evaluate acceptability, feasibility and sustainability of the approach for longer-term and larger-scale deployment. Identifying new, sustainable ways to improve dietary quality and reduce stunting would help to accelerate progress towards 2030 global targets, and could have major benefits for long-term health, development and human capital.

**Author affiliations**
¹Public and Ecosystem Health, Cornell University, Ithaca, New York, USA
²Nutrition, Zvitambo Institute for Maternal and Child Health Research, Harare, Zimbabwe
³Nutrition, Ministry of Health and Child Care, Harare, Zimbabwe
⁴Blizard Institute, Queen Mary University, London, UK
⁵CIMMYT, Harare, Zimbabwe
⁶Queen Mary University, London, UK
⁷Zvitambo Institute for Maternal and Child Health Research, Harare, Zimbabwe
⁸University of Southampton Faculty of Medicine, Southampton, UK
⁹Barts and The London School of Medicine, London, UK
¹⁰Biostatistics & IT, Zvitambo Institute, Harare, Zimbabwe

**Acknowledgements** We appreciate the families in the formative study and the Shurugwi Community Engagement Advisory Board who helped in the design and conduct of the trial.

**Contributors** Trial design: LES, CB, RR, NVT, DTC, JC, TN, TB, KD, KM, JS, PK, RN and AP. Secured funding: LES, CB, RR, NVT, JC, TN, TB, KD, KM, JS, PK, RN and AP. Agriculture design and expertise: JC and TN. Laboratory design and methods development: CB, RR, KM, JS, PK and AP. Data management and analysis plan: BM, BC and RN. Formative work: LES, DTC, SF, NVT, LL, TB, KD, BM, DC and AP. Qualitative design and expertise: LES, TB, KD, DTC, SF, EM, NVT, LL and MM. Training and implementation: DTC, AT, BM and KM. Study oversight: LES, JC, LL, DTC, RN and AP. All authors read and approved the final manuscript.

**Funding** UK Research and Innovation Biotechnology and Biological Sciences Research Council (UKRI-BBSRC), from the Global Challenges Research Fund (GCRF) under Food & Nutrition Research for Health in the Developing World: Bioavailability and Nutrient Content. RR and AP are funded by Wellcome (grants 206455/Z/17/Z and 108065/Z/15/Z). CB is funded by a joint award from Wellcome and the Royal Society (206225/Z/17/Z). The second qualitative substudy is funded by the Arts and Humanities Research Council (AHRC), UK (AH/T004428/1).

**Disclaimer** Funding bodies had no role in the study design, implementation, analysis and interpretation of the data. The principal investigators had no financial and other competing interests.

**Competing interests** None declared.

**Patient and public involvement** Patients and/or the public were not involved in the design, or conduct, or reporting, or dissemination plans of this research.

**Patient consent for publication** Not applicable.

**Provenance and peer review** Not commissioned; externally peer reviewed.

**ORCID iDs**
Dexter. T Chagwena http://orcid.org/0000-0001-7985-2924
Ruairi Robertson http://orcid.org/0000-0002-3719-2056
Paul Kelly http://orcid.org/0000-0003-0844-6448
Robert Ntozini http://orcid.org/0000-0002-8543-2835

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
