## [Reviewer comments · BMJ Open]

ARTICLE DETAILS

TITLE (PROVISIONAL)	Child Health, Agriculture and Integrated Nutrition (CHAIN): protocol for a randomized controlled trial of improved infant and young child feeding in rural Zimbabwe
AUTHORS	Smith, Laura; Chagwena, Dexter; Bourke, Claire; Robertson, Ruairi; Fernando, Shamiso; Tavengwa, Naume; Cairns, Jill; Ndhlela, Thokozile; Matumbu, Exhibit; Brown, Tim; Datta, Kavita; Mutasa, Batsirai; Tengende, Alice; Chidhanguro, Dzivaizdo; Langhaug, Lisa; Makanza, Maggie; Chasekwa, Bernard; Mutasa, Kuda; Swann, Jonathan; Kelly, Paul; Ntozini, Robert; Prendergast, Andrew

VERSION 1 – REVIEW

REVIEWER	Jeyakumar, Angeline Savithribai Phule University of Pune, Interdisciplinary school of health sciences
REVIEW RETURNED	04-Jan-2022

GENERAL COMMENTS	The laboratory component requires review from an expert in that field. The research team's effort to address stunting among children through IYCF practices and complementary feeding using locally available foods is highly appreciated. The following points needed to be addressed before proceeding with the trial:  • Is the trial an individual randomised control trial or individual randomised group trial? • In the research rationale, interventions addressing the risk factors in isolation has been cited as barriers to uptake, how do we know that his multicomponent strategy will be feasible to administer and uptake? • Locally grown foods have been chosen for sustainability. The team suggests household level agriculture as a sustainable intervention. Is this a practical approach? This is a long-term strategy. Does the trial offer short and medium term solutions to address stunting? • PVA maize improved vitamin A levels. There is only one citation to prove this. Similar with other foods such as sugar beans. More references can be added to support the evidence of benefits. • In study setting, it is mentioned that the CHAIN trail is being conducted, how far the trail
---

	has progressed? This needs to be clearly mentioned.  • Details of administering feeding supplements are not clear in the protocol.  • How much of the supplements will be distributed to the caretakers? • Weight of the food consumed by the child will vary. How will the intake be monitored? • What will be the duration of each session in the BCC? • What will be the medium of communication? • Has the team tested the awareness creation component for its content? • Will there be a pre and post-test for the awareness component? • Are the follow-ups weekly or monthly? • The energy intake will be compared with two standards? WHO and a higher reference (Butt). A higher reference is likely to categorise many to have low energy intake. • Sample size for EED, Immune function and other laboratory tests are not clear. • In sample size (line 24) cause of infant deaths needs to be mentioned for ethical concerns, as it could be an adverse outcome of the intervention. • The RCT is to study the effect of the intervention to prevent stunting, this has been explained in the introduction and conclusion. However, the primary objective here is energy intake. Why not length for age? Why it is mentioned a secondary outcome? Secondary outcomes are additional outcomes monitored to help interpret the results of the primary outcome.
--	--

REVIEWER	Joy, Edward London School of Hygiene and Tropical Medicine, Department of Population Health
REVIEW RETURNED	14-Jan-2022

GENERAL COMMENTS	A successful CHAIN trial will generate valuable information on improved complementary feeding for child growth. The age of children involved in the CHAIN trial falls within the critical window of the first 1000 days when we know that linear child growth often falters in rural, low-income settings, with life-long consequences for health and development. Generally, the manuscript adequately explains the trial protocol. I note that the trial is now complete and my review aims to ensure that the protocol is adequately detailed to provide transparency and replicability. Hopefully this will make it easier to report and interpret the results appropriately. Comments Page 8, line 3. Do the percentages with of inadequate dietary intakes refer to the endline measurements of the 'control' and 'intervention' groups of the SHINE trial? Can the authors clarify, and also indicate whether this population is likely to be ~equivalent to the baseline status of the CHAIN participants? Page 8, line 3. The prevalence of inadequate dietary energy intakes seems very high, is there a chance that dietary energy intakes were systematically under-estimated? Are there other corroborating data from the same area/a similar setting that also indicate a high prevalence of inadequate energy intakes? Page 8, line 22. I realise these are just examples of factors influencing food security, but I suggest adding 'household purchasing power' since it is so influential. Page 9, Study overview. I strongly suggest the authors include a SPIRIT checklist for reporting of trial protocols in the supplementary materials. Many of my comments below relate to missing information according to the SPIRIT checklist. Page 9, line 17. Clarify what is meant by "could ultimately be sustainable through agriculture"? Is this modifications and adaptations to existing local agriculture systems? Presumably the 'local' element is important? Also, presumably it doesn't rely on each household adopting new/modifying their agricultural practices, since they could trade or purchase the foods. Page 9, line 17. It appears that children could be recruited and receive the intervention but then be too old for the follow-up data collection? Please clarify. Page 9, line 38. The jump between efficacy and potential self-sufficiency is quite large! What about e.g.: if efficacious, the trial would provide strong proof-of-principle that a comprehensive improvement to complementary feeding using locally-available foods can substantially improve child nutrition and reduce risk of linear growth failure. Page 10, line 10. In objectives 2, 3 and 4, is there also a window of ages when this is measured, like the primary objective? Page 11, line 5. Please clarify, 'improved dietary intake' of what? Page 11, line 12. Are all forms of migration included? Rural-urban, rural-rural, international?
--

	Page 13, line 42. Presumably households could purchase these foods or receive from a public distribution scheme, rather than necessarily having to produce these foods? Page 14, line 40. Were households that participated in the formative research eligible for inclusion in the trial? Page 14, line 47. Similarly, what about households that participated in SHINE, were they eligible for CHAIN? Were the trials conducted in the same community and might that have affected who was recruited for CHAIN? If there was overlap between the formative study area or SHINE trial with participants in the CHAIN trial, then the authors will need to explain whether these households are likely to be representative of the wider population and any risk of bias that may be introduced. Page 14, line 54. Consider moving the information on sensitisation/referral for recruitment and the brief summary of the trial schedule to other sections, or deleting if it's covered elsewhere. Page 14, line 54. Were there also sensitisation activities with the wider community, so they could understand what was happening with their neighbours/community members? Page 14, line 56. Just to clarify, there was no random selection of participating households from a pool of potentially eligible households? Page 14, line 58. There should be a brief description of the training that research nurses (and VHW) completed. Page 14, line 58. Presumably written consent was sought from the parent/legal guardian on behalf of the participant child? Please clarify. Page 15, line 15. Specify the primary outcome unit of measurement Page 15, line 24. The authors could consider having 'household level' inclusion criteria (e.g. planning to live in the study area for the duration of the trial), and 'individual level' inclusion criteria. Regarding the criteria on planning to live in the study area, information on how this was assessed will be required. Page 15, line 36. What about other disabilities that might affect physical activity and energy requirements? Page 15, line 45. Please include dates in the trial schedule, including for the initiation and completion of recruitment. Page 15, line 49. What was the protocol if the potentially-eligible child and respective parent/guardian were not available for screening/informed consent? Were revisits arranged? Page 15, line 49. Was there any assessment of the wider household's willingness to participate? Page 15, line 49. From the description of recruitment, it sounds like there was no random selection of households from a population of potentially-eligible households. Can the authors indicate what proportion of potentially-eligible households (from the VHW registers) were recruited? How did research nurses decide which households to visit? Did recruitment stop simply when the target
--	---

	sample size was reached? Page 15, line 56. Were participants referred to health clinics if SAM/MAM were indicated, like they were at endline? Page 15, line 56. Details of all measurement instruments are required. Page 16, line 5. What was the protocol for getting the sample, finger prick? Were participants referred to health clinics if anaemia was indicated at baseline, like they were at endline? Page 16, line 12. Specify the method of generating the allocation sequence. Also, it's not clear how this was applied. Were household IDs allocated to treatment arms and at what point following recruitment? Page 16, line 19. Will only one of the twins be recruited? Presumably yes, otherwise that could introduce some bias (albeit likely a very rare occurrence) as they are not independent of each other. Also, recruiting both twins might allow identification of the household. If both were recruited, how were these potential issues handled? Page 16, line 26. Clarify how data and laboratory analysts are blinded to allocated treatment arm, i.e. the specific aspects of participant and sample labelling protocols and access to linked datasets? Page 16, line 26. Did the authors plan for any circumstances under which unblinding was permissible? Page 17, line 13. Clarify that, if a caregiver moves out of the study area, the respective child participant will be considered lost-to-follow up and that no attempt will be made to retain them. Please also clarify what will be done with their data? Page 17, line 35. Is it the child that receives the biofortified beans or the household? Presumably the intervention delivers to the household, and it is then under household/caregiver control whether or not the child receives it. Page 17, line 40. Consider amending to "...ensure the daily recommended nutrient intake is met if the food supplements are consumed". Page 18, line 10. Are 'intervention nurses' the same as 'research nurses'? And did intervention nurses encourage (not just monitor) adherence to the treatment? Page 18, line 12. How often did monitoring visits occur? Page 18, line 40. Please clarify the estimated length of time exposed to the intervention (might need some brief summary stats, e.g. Q1, median, Q3 and range). Page 18, line 42. What was the protocol if the child participant was not there? Page 18, line 45. Please detail the protocols for collection of blood, urine and stool samples, including the instruments/materials used. E.g. important to know the volume of blood collected, was it collected into a particular type of vacutainer, was it centrifuged in the
--	--

	field, what temperature was it stored at, etc.? Page 19, line 1. Estimating dietary energy (and nutrient) intakes requires matching of consumption to composition data. Methods for this need inclusion. Page 19, line 5. How will the two datapoints be used for the subset with two dietary estimates? Will the NCI method be used to estimate usual intakes? Please clarify. Page 19, line 10. Please provide relevant references for the statement “This method provides a robust and validated measure of nutrient intake based on a comprehensive and standardized assessment”. Page 19, line 17. How will recipe data be handled? Page 19, line 47. Replace ‘actual’ with ‘estimated’. Page 19, line 47. (This might apply to inclusion/exclusion criteria as well) How were exclusively breastfed children handled? Page 19, line 56. How is the average intake of breastmilk estimated? Page 20, line 47. Please consider providing the data collection forms as supplementary materials. Page 20, line 49. Were there any data validation steps during data collection, e.g. daily checks by a field supervisor/data manager? Page 21, line 47. For how long will data be stored? Page 21, line 45. There’s a good chance there will be some implausible or outlying values of estimated daily energy intake. How will these be identified and handled? Page 21, line 58. Will group means be compared? Page 22, line 8. Will there be any adjustments for multiple comparisons? Page 22, line 10. Which adherence data will be used? Not clear what the metric will be. Page 22, line 15. How is maternal HIV status known? Was this covered by the informed consent process? Page 22, line 22. Please include a section on sample management, e.g. how will samples be identified, how will they be handled in the field, what is the aliquoting process, where will they be stored, how will they be shipped to analysis labs, any management of sample lists/cross validation for quality checks, how long will they be stored for after analysis, etc.? Page 24, line 40. The qualitative studies will be conducted on subsets of the main trial participants, and will target certain groups. This could potentially allow the identification of participants. Please can the authors include information on how confidentiality will be maintained, including how to prevent participant identifiers from the qualitative study being used to identify biological data. Page 25, line 20. Can information be generated on the cost of the inputs and time it takes to implement? Important for considering
--	--

	costings and feasibility of future interventions. Page 26, line 45. Is the independent safety monitor in place of a Data Monitoring Committee? Please clarify the role of the independent safety monitor and what data they review, since outcome data are only collected at endline. Page 26, line 57. Was there any training on COVID mitigation protocols provided to research staff? Page 27, line 15. Was any compensation provided to trial participants/their caregivers? Page 27, line 18. Please include the ethics application reference number. Page 27, line 20. Please include contact details of the trial sponsor. Page 27, line 20. Was it only mothers who could provide consent on behalf of the participants? What about fathers or legal guardians? Page 27, line 22. Clarify this was consent provided on behalf of the child participant. Also, model consent forms should be included in supplementary materials. Page 27, line 29. Were parents/guardians given time to consider participation? Page 27, line 34. Please clarify the implications of the statement "Mothers aged between 15-18 years are considered emancipated minors under Zimbabwean law"? Does that mean children of these mothers were considered eligible for the trial, and these young mothers were considered able to provide consent in the same way that mothers >18 years were? Page 29, line 12. Presumably the authors consider the intervention a "proof of concept"? Or do the authors consider that the intervention could be rolled out more or less as it looks like in the trial, with food rations distributed directly to households? Page 29, line 30. Clarify that it was mothers (or parents/guardians?) providing consent on behalf of their children. Table 2. For the outcome EED, the trial registration mentions use of principal component analysis, but this is not mentioned here. Please clarify. Table 3. The schedule would be more useful if it specified the length of time for each stage, especially the intervention phase. Table 6. Some analytes are not trial outcomes. How will these data be handled and reported?
--	--

VERSION 1 – AUTHOR RESPONSE

Reviewer 1
Comments

The research team’s effort to address stunting among children through IYCF practices and complementary feeding using locally available foods is highly appreciated.

The following points needed to be addressed before proceeding with the trial:

- Is the trial an individual randomised control trial or individual randomised group trial?

Response: Indicated in the methods that it is an individually randomized households trial.

- In the research rationale, interventions addressing the risk factors in isolation has been cited as barriers to uptake, how do we know that this multicomponent strategy will be feasible to administer and uptake?

Response: We have presented the formative work and indicated how we tested feasibility and uptake of this multicomponent IYCF, and behaviour change strategy.

- Locally grown foods have been chosen for sustainability. The team suggests household level agriculture as a sustainable intervention. Is this a practical approach? This is a long-term strategy. Does the trial offer short- and medium-term solutions to address stunting?

Response: We have indicated that our intervention addresses the challenge of nutrient-gap among young children during the complementary feeding period of 6 to 23 months. This is an age group that is particularly vulnerable and at increased risk of malnutrition according to Zimbabwean and other resource-limited countries literature. This contributes to addressing stunting. Sustainability is long-term, these products are available on the local market. The solution of this trial is to meet nutrient requirements which is a short-term solution to address stunting.

We chose HH level as the unit so that we could run an efficient individually randomized trial, as this trial is a proof of concept to address nutrient gap. In Zimbabwe is sparsely populated and for the trial this was an efficient way to run a trial.

From literature group level agric intervention and our formative work shows that it is the most efficient way to intervene. If our trial is successful, we would propose group level agriculture approach for sustainability and effectiveness.

The individual randomized trial was the most effective design to test our hypothesis compared to a cluster randomized trial that would require a larger sample size. This trial focus on a solution that is a short-term solution to addressing stunting.

- PVA maize improved vitamin A levels. There is only one citation to prove this. Similar with other foods such as sugar beans. More references can be added to support the evidence of benefits.

Response: Included additional references from the Zambian experience which is a similar setting to the study site.

- In study setting, it is mentioned that the CHAIN trial is being conducted, how far the trial has progressed? This needs to be clearly mentioned.

Response: Addressed by providing the study commencement date of recruitment and study participants follow up period.

Recruitment of study participants was commenced on 26 April 2021 and study participants followed until March 2022. The trial implementation was affected by the Covid19 pandemic.

- Details of administering feeding supplements are not clear in the protocol. • How much of the supplements will be distributed to the caretakers?

Response: This was addressed as text under Table 5.

Food supplements will be delivered monthly by VHWs. 1Mealie Meal (PVA maize) will be provided in 500g bags, with 3 bags/month (1500g) between 6-8 months of age and 5 bags/month (2500g) between 9-11 months of age. Households in the IYCF arm will receive the same amount of mealie meal per month 2SQ-LNS will be supplied monthly to ensure 1 x 20g sachet per day can be provided (30 or 31 sachets per month). 3Whole egg powder will be delivered as a 500g bag per month.

4Moringa leaf powder will be supplied in 175g bags, with 1 bag/month (175g) between 6-8 months of age and 2 bags/month (350g) between 9-11 months of age. 5SNUA 45 sugar bean powder s will be supplied in 175g bags, with 1 bag/month (175g) between 6-8 months of age and 2 bags/month (375g) between 9-11 months of age. These quantities allow for 15% extra in case of spillage or sharing

- Weight of the food consumed by the child will vary. How will the intake be monitored?

Daily intake will not be monitored but measured through a 24 hour multiple pass dietary recall as indicated under Follow-Up Data Collection in the methodology section.

- What will be the duration of each session in the BCC?

Response: Addressed as text supplementing Table 4

Each module session will be delivered for approximately 60 minutes. If a module is not delivered within the intervention window (i.e. appropriate infant age), the VHW will try to catch up by scheduling a new date as soon as possible. Each module will be delivered to the mother and her family. If the rescheduled module for modules 1.0 and 2.0 overlap, these two modules will be delivered at the same time. If the rescheduled module overlaps with the next visit for other modules (2.0, 2.1, 2.2, 2.3 etc) the visits will be scheduled at least 3 days apart so that families have time to absorb the new material. Delivery of IYCF-plus modules has therefore been designed to be flexible following complementary feeding guidance. Experience from formative work showed that it is feasible to deliver the combined modules at once.

- What will be the medium of communication?

Response: Interpersonal face-to-face counseling sessions as indicated under Intervention Delivery

Behavioural modules: A total of nine interpersonal face-to-face counseling session modules will be delivered to caregivers in each arm by VHWs during 10 home visits, which coincide with key infant ages, so that sequential age-appropriate messages about complementary feeding are introduced and reinforced (Table 4)

- Has the team tested the awareness creation component for its content?

Response: Addressed under formative research work. The formative paper was referenced as unpublished work in preparation.

The formative work also tested feasibility and uptake of this multicomponent complementary feeding and behaviour change strategy among similar rural households to the study setting.

- Will there be a pre and post-test for the awareness component?

Response: Qualitative interviews explored message awareness in a subset of households. We did not conduct post test awareness but we did measure self-adherence to the promoted practices during the formative work.

- Are the follow-ups weekly or monthly?

Response: Indicated weekly and monthly under the methodology.

Intervention follow up were weekly in the first month with introduction of new foods and then monthly from 7 months of age onwards.

- The energy intake will be compared with two standards? WHO and a higher reference (Butte). A higher reference is likely to categorise many to have low energy intake.

Response: We clarified this in the protocol paper where we are using the method from Butte on estimating emerging requirements for children low-income countries.

Actual energy and nutrient intakes will be derived from the baby weight, dietary recall and breast milk intake at 9 months. The energy intake from breast milk will be estimated using the estimated nutrient in 550g based on the study findings (Michaelsen; 2000). The calculation of energy required from food uses estimates from Butte for infants in low-income settings due to the greater infection burden(Butte;2005). The means and standard deviations for each trial arm will be calculated for energy balance.

- Sample size for EED, Immune function and other laboratory tests are not clear.

Response: This is a tertiary outcome being measured on everyone with samples available for baseline and endline. The study is not powered for this outcome and is exploratory.

- In sample size (line 24) cause of infant deaths needs to be mentioned for ethical concerns, as it could be an adverse outcome of the intervention.

Response: The sample size of 192 infants assumes 10% loss to follow-up due to withdrawal and infant deaths, meaning there will be an estimated 86 evaluable infants per group at endline. Infant deaths at this stage are rare but all causes of deaths will be reported as adverse events.

- The RCT is to study the effect of the intervention to prevent stunting, this has been explained in the introduction and conclusion. However, the primary objective here is energy intake. Why not length for age? Why it is mentioned a secondary outcome? Secondary outcomes are additional outcomes monitored to help interpret the results of the primary outcome.

Response: RCT is to study the effect of the intervention on nutrient intake. This is a one known factor to address stunting, because stunting is multifaceted commencing in utero continuing to complementary

Stunting was not in the scope of this trial. This was a short proof of concept trial targeting 6 to 12 months children. If we are to prove that our intervention works to improve nutrient intake, we would propose

Our trial was not powered enough to report on stunting as a primary outcome, hence we included length for age as a secondary outcome.

Reviewer 2

A successful CHAIN trial will generate valuable information on improved complementary feeding for child growth. The age of children involved in the CHAIN trial falls within the critical window of the first 1000 days when we know that linear child growth often falters in rural, low-income settings, with life-long consequences for health and development. Generally, the manuscript adequately explains the trial protocol. I note that the trial is now complete and my review aims to ensure that the protocol is adequately detailed to provide transparency and replicability. Hopefully this will make it easier to report and interpret the results appropriately.

Comments

Page 8, line 3. Do the percentages with of inadequate dietary intakes refer to the endline measurements of the 'control' and 'intervention' groups of the SHINE trial? Can the authors clarify, and also indicate whether this population is likely to be ~equivalent to the baseline status of the CHAIN participants?

Response: Yes, the percentages refer to endline dietary intake. We expect the CHAIN population to be similar to the SHINE population described above as they are from the same study district where SHINE was conducted up to 2017. This has been clarified.

Page 8, line 3. The prevalence of inadequate dietary energy intakes seems very high, is there a chance that dietary energy intakes were systematically underestimated? Are there other corroborating data from the same area/a similar setting that also indicate a high prevalence of inadequate energy intakes?

Response: These numbers were checked. For energy, we have adjusted the value after reviewing energy requirements. Folate and iron remain the same. The requirement for folate increases at 12 months of age and diets with SQ-LNS do not contain enough to meet the requirement. Our prior work in the area has indicated similar deficiencies.

Page 8, line 22. I realise these are just examples of factors influencing food security, but I suggest adding 'household purchasing power' since it is so influential.

Response: This has been added.

Page 9, Study overview. I strongly suggest the authors include a SPIRIT checklist for reporting of trial protocols in the supplementary materials. Many of my comments below relate to missing information according to the SPIRIT checklist.

Response: This has been included in supplementary materials

Page 9, line 17. Clarify what is meant by "could ultimately be sustainable through agriculture"? Is this modifications and adaptations to existing local agriculture systems? Presumably the 'local' element is important? Also, presumably it doesn't rely on each household adopting new/modifying their agricultural practices, since they could trade or purchase the foods.

Response: This has been clarified.

Page 9, line 17. It appears that children could be recruited and receive the intervention but then be too old for the follow-up data collection? Please clarify.

Response: This has been clarified.

Infants will be enrolled between 5-6 months of age, and begin receiving the intervention at 5 months of age. The endline visit is conducted at 9 months of age, but has an allowable visit window from 9 months up until the child turns 12 months of age, to maximize endline data collection.

Page 9, line 38. The jump between efficacy and potential self-sufficiency is quite large! What about e.g.: if efficacious, the trial would provide strong proof-of-principle that a comprehensive improvement to complementary feeding using locally-available foods can substantially improve child nutrition and reduce risk of linear growth failure.

Response: This has been modified as suggested.

Page 10, line 10. In objectives 2, 3 and 4, is there also a window of ages when this is measured, like the primary objective?

Response: Yes, this has been added.

Page 11, line 5. Please clarify, 'improved dietary intake' of what?

Response: This has been clarified to indicate macro- and micronutrient intake. As this aim is exploratory, we will look at all nutrients.

Page 11, line 12. Are all forms of migration included? Rural-urban, rural-rural, international?

Response: Yes, all forms of migration are included. This has been added.

Page 13, line 42. Presumably households could purchase these foods or receive from a public distribution scheme, rather than necessarily having to produce these foods?

Response: Yes, there is also good potential for local small and medium enterprises to produce and process these products. This has been added.

Page 14, line 40. Were households that participated in the formative research eligible for inclusion in the trial?

Response: Yes, the households who participated in the formative research were eligible for inclusion if they had a subsequent child aged 5-6 months during the recruitment period, since this was not a stated exclusion criterion.

Page 14, line 47. Similarly, what about households that participated in SHINE, were they eligible for CHAIN? Were the trials conducted in the same community and might that have affected who was recruited for CHAIN? If there was overlap between the formative study area or SHINE trial with participants in the CHAIN trial, then the authors will need to explain whether these households are likely to be representative of the wider population and any risk of bias that may be introduced.

Response: Yes, the households who participated in the SHINE trial were eligible for inclusion if they had a subsequent child aged 5-6 months during the recruitment period, as being in a previous trial was not an exclusion criterion. The SHINE trial recruited infants between 2012-2015 and only half of the participants received any sort of nutrition intervention.

The CHAIN formative research was conducted with infants 6-18 months at one clinic in the study district one year prior to CHAIN enrolment. It is possible that households could have participated in a previous research trial, but all households were selected from the same source population, had an equal chance of participation and were randomized between the two arms limiting the potential for selection bias. Only one family had participated in the CHAIN formative study with a previous child.

Page 14, line 54. Consider moving the information on sensitisation/referral for recruitment and the brief summary of the trial schedule to other sections, or deleting if it's covered elsewhere.

Response: This has been done.

Page 14, line 54. Were there also sensitisation activities with the wider community, so they could understand what was happening with their neighbours/community members?

Response: Yes, sensitization was also conducted with community stakeholders.

Page 14, line 56. Just to clarify, there was no random selection of participating households from a pool of potentially eligible households?

Response: We identified all eligible households who had a child between 5-6 months during the recruitment period. There were not enough eligible households to subselect a pool from the eligible households.

Page 14, line 58. There should be a brief description of the training that research nurses (and VHW) completed.

Response: This has been added.

Page 14, line 58. Presumably written consent was sought from the parent/legal guardian on behalf of the participant child? Please clarify.

Response: Correct. This has been added.

Page 15, line 15. Specify the primary outcome unit of measurement

Response: This has been added.

Page 15, line 24. The authors could consider having 'household level' inclusion criteria (e.g. planning to live in the study area for the duration of the trial), and 'individual level' inclusion criteria.

Regarding the criteria on planning to live in the study area, information on how this was assessed will be required.

Response: This has been added.

Page 15, line 36. What about other disabilities that might affect physical activity and energy requirements?

Response: The only exclusion criterion regarding disabilities was a disability that interferes with feeding. We did not screen for chronic disease which may affect energy requirements.

Page 15, line 45. Please include dates in the trial schedule, including for the initiation and completion of recruitment.

Response: This has been added in the section Study Setting and Recruitment and in a footnote in Table 3.

Page 15, line 49. What was the protocol if the potentially-eligible child and respective parent/guardian were not available for screening/informed consent? Were revisits arranged?

Response: This has been added.

Page 15, line 49. Was there any assessment of the wider household's willingness to participate?

Response: This has been added. All household members were encouraged to be present for consent and subsequent intervention and research visits.

Page 15, line 49. From the description of recruitment, it sounds like there was no random selection of households from a population of potentially-eligible households. Can the authors indicate what proportion of potentially-eligible households (from the VHW registers) were recruited? How did research nurses decide which households to visit? Did recruitment stop simply when the target sample size was reached?

Response: This has been added. 282 infants were identified through VHW registers who would become 5 months of age during the enrolment period. Consent visits were scheduled as close to children turning 5 months as possible and continued until the required sample size was reached. There was no random selection within an eligible pool.

Page 15, line 56. Were participants referred to health clinics if SAM/MAM were indicated, like they were at endline?

Response: Yes, referral also occurred during the baseline visit and this has been added. Children with symptomatic mild to moderate anaemia (<11 g/dL) or with severe anaemia (<7 g/dL) were referred to local clinics. Children with moderate or severe acute malnutrition (MUAC<125mm, or weight-for-length Z-score <-2) were also referred to local clinics.

Page 15, line 56. Details of all measurement instruments are required.

Response: This has been added. Maternal and infant height (ShorrBoard®) infant, child, adult measur, weight (Seca 874DR Mother-Baby scale) and head and mid-upper arm circumference ShoreTape®) will be measured.

Page 16, line 5. What was the protocol for getting the sample, finger prick? Were participants referred to health clinics if anaemia was indicated at baseline, like they were at endline?

Response: This has been added.

Infant blood was collected using a butterfly or toddler Tenderfoot® device, with a drop of blood put into the HemoCue at the household. Referral for anemia was done and this has also been added. Children with symptomatic mild to moderate anaemia (<11 g/dL) or with severe anaemia (<7 g/dL) were referred to local clinics.

Page 16, line 12. Specify the method of generating the allocation sequence. Also, it's not clear how this was applied. Were household IDs allocated to treatment arms and at what point following recruitment?

Response: This has been added. The randomization schema was pre-prepared by the trial statistician using the RALLOC command in STATA 14, using random permuted blocks, with a 1:1 allocation to IYCF or IYCF-plus. Household IDs were pre-generated and allocated to treatment arms prior to recruitment into the study. Household IDs were assigned to a specific household after consent.

Page 16, line 19. Will only one of the twins be recruited? Presumably yes, otherwise that could introduce some bias (albeit likely a very rare occurrence) as they are not independent of each other. Also, recruiting both twins might allow identification of the household. If both were recruited, how were these potential issues handled?

Response: Both twins in eligible households were recruited, because it would be difficult to enrol one infant and not the other into a study. A sensitivity analysis will be conducted excluding one twin.

Page 16, line 26. Clarify how data and laboratory analysts are blinded to allocated treatment arm, i.e. the specific aspects of participant and sample labelling protocols and access to linked datasets?

Response: This has been added. All samples and trial records are identified by the participant ID number which does not include arm allocation. All laboratory analyses and data analyses are conducted irrespective of trial arm and then merged by the trial statistician before reporting.

Page 16, line 26. Did the authors plan for any circumstances under which unblinding was permissible?

Response: Monthly reports were provided to the trial monitor and MRCZ with adverse events reported by trial arm. No other unblinding was provisioned for.

Page 17, line 13. Clarify that, if a caregiver moves out of the study area, the respective child participant will be considered lost-to-follow up and that no attempt will be made to retain them. Please also clarify what will be done with their data?

Response: Endline data collection visits will be conducted regardless of where the caregiver moves to, but the interventions would not be delivered. This ensures that we collect complete endline data for our intention-to-treat analyses. This has been clarified.

Page 17, line 35. Is it the child that receives the biofortified beans or the household? Presumably the intervention delivers to the household, and it is then under household/caregiver control whether or not the child receives it.

Response: Families receive NUA-45 biofortified bean powder, whole egg powder and moringa leaf powder for provision to the study child. This has been clarified.

Page 17, line 40. Consider amending to "...ensure the daily recommended nutrient intake is met if the food supplements are consumed".

Response: This has been modified.

Page 18, line 10. Are 'intervention nurses' the same as 'research nurses'? And did intervention nurses encourage (not just monitor) adherence to the treatment?

Response: Intervention nurses were a separate cadre from research nurses. Intervention nurses did not provide counselling to mothers but did provide supportive supervision to VHWs by scheduled attendance at some household visits to provide feedback and by conducting unscheduled spot checks. This has been clarified.

Page 18, line 12. How often did monitoring visits occur?

Response: Intervention nurses attended visits each time a VHW was delivering a module for the first time and additionally if needed. This has been added.

Page 18, line 40. Please clarify the estimated length of time exposed to the intervention (might need some brief summary stats, e.g. Q1, median, Q3 and range).

Response: This is clarified

Page 18, line 42. What was the protocol if the child participant was not there?

Response: If the child was not present, the visit was rescheduled.

Page 18, line 45. Please detail the protocols for collection of blood, urine and stool samples, including the instruments/materials used. E.g. important to know the volume of blood collected, was it collected into a particular type of vacutainer, was it centrifuged in the field, what temperature was it stored at, etc.?

Response: This has been added.

Page 19, line 1. Estimating dietary energy (and nutrient) intakes requires matching of consumption to composition data. Methods for this need inclusion.

Response: This has been added. Data from the 24-hour recall will be converted to observed energy and nutrient intakes by the following steps:

1. Ingredients and portion sizes were measured and weighed in grams where possible. For ingredients that could not be weighed, they will be converted to grams using locally collected data on food densities, supplemented with food density data from the FAO, USDA and NDSR ³⁴⁻³⁶.
2. Mixed dishes were disaggregated into ingredients and entered into Nutrisurvey to calculate nutrients in 100 grams of food.

3. Energy from each individual food/ingredient will be estimated using food composition data from regional food databases and USDA databases that have been collated for use in Zimbabwe over several studies.
4. Total energy (and nutrient) intake is estimated from the child's daily food intake.

Page 19, line 5. How will the two datapoint be used for the subset with two dietary estimates? Will the NCI method be used to estimate usual intakes? Please clarify.

Response: This has been added.

The primary outcome, percent meeting energy intake will be calculated in two ways. First, we will use measured intake data from 1 dietary recall from all participants. Second, we will use the NCI method for calculated usual overall average intake. We will estimate the mean and percentiles of usual energy intake distributions using the National Cancer Institute (NCI) method[26, 27]. This method adjusts for measurement error – primarily due to day-to-day variability in intakes – in observed, single-day estimates of nutrient intakes.

We will use the NCI macros, DISTRIB and MIXTRAN, plus bootstrapped standard errors for hypothesis testing [25, 27]. The MIXTRAN macro fits a mixed effects model of usual energy intake, and the DISTRIB macro uses a Monte Carlo procedure to estimate percentiles of the usual intake distribution.

We will use the MIXTRAN macro to fit a model of energy intake with a fixed effect for intervention group and the DISTRIB macro to estimate usual energy intake distributions by treatment group. We will bootstrap the parameters to estimate the standard error of the mean usual intakes. We will use the point estimates and standard error estimates to construct 95% confidence intervals and calculate p-values for difference in mean by intervention group based on Welch's t-test.

Bootstrapped standard errors will be calculated using bootstrap samples. When intake or the number of repeat recalls is low, it is common for the NCI method to fail to converge. In cases where convergence prohibits bootstrapping, we will test hypotheses using the Wilcoxon testing for clustered method.

Page 19, line 10. Please provide relevant references for the statement "This method provides a robust and validated measure of nutrient intake based on a comprehensive and standardized assessment".

Response: This has been added.

Page 19, line 17. How will recipe data be handled?

Response: This has been added. Mixed dishes were disaggregated into ingredients and entered into Nutrisurvey to calculate nutrients in 100 grams of food.

Page 19, line 47. Replace 'actual' with 'estimated'.

Response: Corrected.

Page 19, line 47. (This might apply to inclusion/exclusion criteria as well) How were exclusively breastfed children handled?

Response: No children were exclusively breastfed during the study, since children started the intervention at 6 months of age. Intervention materials addressed introduction of complementary foods.

Page 19, line 56. How is the average intake of breastmilk estimated?

Response: For breastfeeding children, we will calculate the required nutrient intake from complementary foods by subtracting the amount of each nutrient in 550 g breast milk from the total requirement which is the estimated intake of breast milk for 9-11 month old children. This has been clarified.

Page 20, line 47. Please consider providing the data collection forms as supplementary materials.

Response: These will be published online at Open Science Framework.

Page 20, line 49. Were there any data validation steps during data collection, e.g. daily checks by a field supervisor/data manager?

Response: All data were checked daily by the field data officer, and implausible values are verified or recollected.

Page 21, line 47. For how long will data be stored?

Response: Data will be stored for 20 years, as per the trial protocol and Sponsor requirements. This has been added.

Page 21, line 45. There's a good chance there will be some implausible or outlying values of estimated daily energy intake. How will these be identified and handled?

Response: All data were checked daily in real time by the field data officer, with implausible values verified or recollected. This has been clarified.

Page 21, line 58. Will group means be compared?

Response: This has been added. We will interpret the mean intake of each nutrient for the primary and secondary analysis in each arm, and will present mean differences with the 95% confidence interval for interpretation.

Page 22, line 8. Will there be any adjustments for multiple comparisons?

Response: Yes, we will adjust for multiple comparisons and this is outlined in our SAP that will be published online separately.

Page 22, line 10. Which adherence data will be used? Not clear what the metric will be.

Response: This has been added. Adherence will be assessed by self-report in a 7-day recall.

Page 22, line 15. How is maternal HIV status known? Was this covered by the informed consent process?

Response: This has been added. Maternal HIV status is collected by self-report and review of maternal handheld records. HIV testing was not done by the study team.

Page 22, line 22. Please include a section on sample management, e.g. how will samples be identified, how will they be handled in the field, what is the aliquoting process, where will they be stored, how will they be shipped to analysis labs, any management of sample lists/cross validation for quality checks, how long will they be stored for after analysis, etc.?

Response: This has been added. Preprinted barcodes identifying the participant ID and sample type were adhered to the collection tube in the field and transported in a cooler bag to the lab. When samples arrived at the lab, they were processed and aliquoted into cryovials which were labeled with barcodes identifying the participant ID, sample type and aliquot number. Samples were stored in the field lab at -80° Celsius. At regular intervals, samples were transported to the main lab in Harare, Zimbabwe where they were stored at -80° Celsius until analysis or shipment. All samples will be shipped to external labs on dry ice. Sample lists will be maintained in the main trial database. If participants consented to long-term storage, samples will be stored indefinitely.

Page 24, line 40. The qualitative studies will be conducted on subsets of the main trial participants, and will target certain groups. This could potentially allow the identification of participants. Please can the authors include information on how confidentiality will be maintained, including how to prevent participant identifiers from the qualitative study being used to identify biological data.

Response: The qualitative studies aim to understand different types of experiences within the community. All identifying data will remain confidential as in the larger study.

Page 25, line 20. Can information be generated on the cost of the inputs and time it takes to implement? Important for considering costings and feasibility of future interventions.

Response: We aim to include this information in the main trial paper. For this study, some of the study inputs were more expensive due to the limitations faced in a research study.

Page 26, line 45. Is the independent safety monitor in place of a Data Monitoring Committee? Please clarify the role of the independent safety monitor and what data they review, since outcome data are only collected at endline.

Response: Yes, the data safety monitor was suggested by the local ethical review board (MRCZ) instead of a DMC. She reviewed adverse event data by arm monthly throughout the study and provided an independent opinion as to whether there were any safety concerns.

Page 26, line 57. Was there any training on COVID mitigation protocols provided to research staff?

Response: Yes, all staff were trained in COVID protocols.

Page 27, line 15. Was any compensation provided to trial participants/their caregivers?

Response: Families were provided with a small gift (soap and Vaseline) for each research data collection visit to compensate for time spent in the trial. This has been added.

Page 27, line 18. Please include the ethics application reference number.

Response: This has been added.

Page 27, line 20. Please include contact details of the trial sponsor.

Response: This has been added.

Page 27, line 20. Was it only mothers who could provide consent on behalf of the participants? What about fathers or legal guardians?

Response: The main caregiver/guardian could consent. This has been clarified.

Page 27, line 22. Clarify this was consent provided on behalf of the child participant. Also, model consent forms should be included in supplementary materials.

Response: All caregivers/legal guardians provided written informed consent on behalf of their child. The consent forms will be available online at Open Science Framework (<https://osf.io/njy2a/>).

Page 27, line 29. Were parents/guardians given time to consider participation?

Response: This has been added. If the caregiver wanted to think about participation or consult, the consenting visit was rescheduled to give her time.

Page 27, line 34. Please clarify the implications of the statement “Mothers aged between 15-18 years are considered emancipated minors under Zimbabwean law”? Does that mean children of these mothers were considered eligible for the trial, and these young mothers were considered able to provide consent in the same way that mothers >18 years were?

Response: This has been clarified. Mothers aged between 15-18 years are considered ‘emancipated minors’ under Zimbabwean law, meaning they could consent on behalf of their child because – even though were under adult age – they were considered competent to provide consent since they are mothers.

Page 29, line 12. Presumably the authors consider the intervention a "proof of concept"? Or do the authors consider that the intervention could be rolled out more or less as it looks like in the trial, with food rations distributed directly to households?

Response: Yes, this is considered a proof of concept; this has been clarified.

Page 29, line 30. Clarify that it was mothers (or parents/guardians?) providing consent on behalf of their children.

Response: This has been clarified.

Table 2. For the outcome EED, the trial registration mentions use of principal component analysis, but this is not mentioned here. Please clarify.

Response: The SAP for that sub-analysis will be published separately. Analytical methods are outside of the scope for Table 2. However, the reviewer is correct that we will use PCA analysis as a data reduction step for handling a large number of biomarkers.

Table 3. The schedule would be more useful if it specified the length of time for each stage, especially the intervention phase.

Response: Table 3 outlines the schedule of data collection visits. Each visit lasted 2-4 hours. The length of the intervention visits has been added in table 4.

Table 6. Some analytes are not trial outcomes. How will these data be handled and reported?

Response: These analytes are for exploratory analysis and are not prespecified outcomes.

VERSION 2 – REVIEW

REVIEWER	Joy, Edward London School of Hygiene and Tropical Medicine, Department of Population Health
REVIEW RETURNED	27-Jun-2022

GENERAL COMMENTS	The protocol is much clearer now and I am happy with the author responses. There are three things which the authors could still consider:  1. It's in the past now, but I do note that recruitment began before the final ethical approval was in place. (Or was the August ethics approval date for version 1.4 a modification?) 2. The authors give two ways of calculating/reporting the primary outcome. I suggest the authors chose one of these. Also, the unit could be further clarified to kcal/day 3. I think it would be preferable to include (at random) one of each twin participant, since twins are not independent of each other. I realise this is very unlikely to make a difference to the stats. There is still the potential issue of identifying participants - two individual IDs from one household ID - and great care will be needed to ensure these participants cannot be identified, especially given the potential sensitivity of some information such as caregiver HIV status.
---

VERSION 2 – AUTHOR RESPONSE

Reviewer: 2

Dr. Edward Joy, London School of Hygiene and Tropical Medicine

Comments to the Author:

The protocol is much clearer now and I am happy with the author responses. There are three things which the authors could still consider:

1. It's in the past now, but I do note that recruitment began before the final ethical approval was in place. (Or was the August ethics approval date for version 1.4 a modification?)

Response: This was the final ethical approval following amendments, and this has been clarified in the main text.

2. The authors give two ways of calculating/reporting the primary outcome. I suggest the authors chose one of these. Also, the unit could be further clarified to kcal/day

Response: We revised the data analysis and reporting of the primary analysis to a single method and unit clarified to kcal/day. The second method will only be utilized for sensitivity analysis as described in the main document.

3. I think it would be preferable to include (at random) one of each twin participant, since twins are not independent of each other. I realize this is very unlikely to make a difference to the stats. There is still the potential issue of identifying participants - two individual IDs from one household ID - and great care will be needed to ensure these participants cannot be identified, especially given the potential sensitivity of some information such as caregiver HIV status.

Response: We included the statement on randomizing one of each twin.